# Carbonized paramagnetic complexes of Mn (II) as contrast agents for precise magnetic resonance imaging of sub-millimeter-sized orthotopic tumors

Ruixue Qin[1,2], Shi Li[1,2], Yuwei Qiu[1,2], Yushuo Feng[1], Yaqing Liu[1], Dandan Ding[1], Lihua Xu[1], Xiaoqian Ma[1], Wenjing Sun[1] & Hongmin Chen [1✉]

Paramagnetic complexes containing gadolinium ions have been widely used for magnetic resonance imaging (MRI) in clinic. However, these paramagnetic complexes pose some safety concerns. There is still a demand for the development of stable MRI contrast agents that exhibit higher sensitivity and superior functionality to existing contrast agents. Here, we develop carbonized paramagnetic complexes of manganese (II) (Mn@CCs) to encapsulate $Mn^{2+}$ in sealed carbonized shells with superhigh $r_1$ relaxivity. Compared to the most common clinical contrast agent Magnevist, investigations in vivo demonstrate that the Mn@CCs cross the intact blood-brain barrier of normal health mice with minor metal deposition; preferentially target the glioma tissues distribute homogeneously with high penetration in an intracranial mouse model; delineate clear tumor margins in MRIs of ultrasmall single-nodule brain tumors, and multi-nodular liver tumors. The sensitivity, accuracy and low toxicity offer by Mn@CCs provides new opportunities for early molecular diagnostics and imaging-guided biomedical applications.

[1] State Key Laboratory of Molecular Vaccinology and Molecular Diagnostics & Center for Molecular Imaging and Translational Medicine, School of Public Health, Xiamen University, 361102 Xiamen, China. [2] These authors contributed equally: Ruixue Qin, Shi Li, Yuwei Qiu. ✉email: hchen@xmu.edu.cn

The early diagnosis of cancers is of great importance in clinics[1]. Magnetic resonance imaging (MRI), ultrasound, and positron emission tomography-computed tomography (PET/CT) are most often used to evaluate deep tumors[2–4]. Among these imaging methods, MRI has the advantage of delivering high-resolution images of brain and liver tumors without ionizing radiation and with virtually unlimited tissue penetration depth[5]. The sensitivity is further increased if gadolinium (Gd) contrast agents approved for human use are employed[6]. However, the use of some Gd chelates can result in a rare but severe complication known as nephrogenic systemic fibrosis (NSF) in patients with kidney disease[7]. Additionally, Gd ions can remain in the body even after a prolonged period of time after an MRI scan[8]. In December 2017, the FDA ordered a black box warning on all Gd chelates in the clinic[9,10]. As a result, three Gd-based contrast agents lost the chance to be used in patients with kidney disease. The FDA also mandated that additional animal and clinical studies must be conducted to assess the safety of these agents[10]. Superparamagnetic iron oxides have been clinically approved in the United States and Europe for liver tumor enhancement[11]. Unfortunately, these nanoparticle contrast agents were never commercially launched, and their development was discontinued in recent decades due to safety concerns[12,13] since iron-based contrast agents release free iron ions that induce the ferroptosis pathway[14–16].

In the clinic, it is often difficult to provide accurate tumor diagnoses without contrast-enhanced MRI. As a result, there is a need for clinically applicable alternatives to the current standard MR complex contrast agent that has been used for nearly 30 years. Mn ions possess the second-highest paramagnetic moment of any element, which is expected to have a good enhancement effect on relaxation rates[17]. As an essential human dietary element, Mn also has essential roles in cell biology, and $Mn^{2+}$ can be used for functional brain imaging owing to the ability of $Mn^{2+}$ to enter cells through $Ca^{2+}$ channels. For these reasons, the potential for using manganese at the center of a paramagnetic complex contrast agent has always excited considerable interest. Although strategies have been developed to enhance their sensitivity on their oxides (i.e., MnOx), its structures could be degraded into free Mn ions to achieve significant enhancement of MRI, the significant toxicity from Mn ions via Fenton-like reaction has been widely reported and limited their translational applications[18–20]. Borrowing the idea from the clinical used Gd chelates, Mangafodipir (Teslascan, Mn-DPDP complex) has been used in clinical trials[21]. However, due to poor clinical performance, and concerns over toxicity, Teslascan was withdrawn from the markets[22]. More recently, researchers have taken a key step forward in developing a novel manganese-based complex contrast agent (Mn-PyC3A) that provides comparable tumor contrast enhancement to Gd-based contrast agents[23]. More importantly, the new agent may be safer than Gd-based contrast agents, because Mn from Mn-PyC3A is much more quickly and thoroughly cleared from the body than Gd from even the 'safest' Gd-based contrast agents. Previous investigations by us and others confirmed that carbonized polymerized shells could encapsulate Gd ions to minimize leakage and enhance their magnetic property and demonstrated the potential applications of these shells for tumor imaging and therapy[24–28].

Glioblastoma (GBM), has a median survival of 14 months after diagnosis[29]. One of the major reasons is the lack of an effective tool for detecting glioma at its early stage when treatment is more sensitive. The blood–brain barrier (BBB) prevents 98% of small molecules and all large molecules from entering the brain[30]. Currently available MRI contrast agents for the detection of brain tumors, such as Magnevist (Gd-DTPA), are nonspecific and only detect tumor masses that have significantly damaged the BBB[31].

However, at the early stage of glioma in the clinic, they cannot be enhanced by the MRI contrast agent in most cases because the BBB has not been significantly damaged. To solve the issue of the commercial MRI contrast agents, we herein developed carbonized paramagnetic complexes of manganese (II) (Mn@CCs) using food additives and amino acids as the precursors. Encapsulating $Mn^{2+}$ in sealed carbonized shells achieved superhigh $Mn^{2+}$ stability and $r_1$ relaxivity (22.1 $mM^{-1}s^{-1}$, 9.4 T). Investigations in vivo on normal healthy mice demonstrated that the Mn@CCs efficiently crossed the intact BBB by multiphoton intravital imaging. Compare to Gd-DTPA, three-dimensional (3D) magnetic resonance images and light-sheet fluorescence microscopy (LSFM) imaging clearly rendered that Mn@CCs preferentially target the glioma tissues and distribute homogeneously with high penetration in an intracranial mouse model, thereby delineating clear brain tumor margins and enabling precise MRI of microscopic single-nodule brain tumors (about 1 mm). Notably, Mn@CCs also detected efficiently multinodular liver tumors (each nodule < 1 mm) in an orthotopic hepatic cancer mouse model. More importantly, Mn@CCs were efficiently excreted from the host at 4 h post-injection through renal clearance with minor metal deposition. The high tumor-to-normal tissue ratio (TNR) offered by Mn@CCs provides new opportunities for sensitive early molecular diagnostics and imaging-guided biomedical applications.

## Results and discussion

**Synthesis and characterization of Mn@CCs.** The Mn@CCs were synthesized by mixing manganese gluconate (glucose-Mn) food nutritional additives and L-aspartic acid, followed by hydrothermal treatment and dialysis. Transmission electron microscopy (TEM) imaging revealed that Mn@CCs were monodisperse with a uniform size of less than 5 nm (Fig. 1a). High-resolution TEM (HR-TEM) imaging showed that the Mn@CCs possessed well-resolved lattice fringes with a spacing of 0.32 nm (Fig. 1b), which can be attributed to the (002) lattice fringes of graphene (Supplementary Fig. 1)[27]. Dynamic light scattering (DLS) showed a relatively narrow size distribution with a hydrodynamic diameter of ~11.3 nm (Fig. 1c). Raman spectroscopy indicated two peaks at 1358 and 1560 $cm^{-1}$, corresponding to the disordered structures or defects (D band) and the graphitic carbon domains (G band), respectively[32]. The intensity ratio ($I_D/I_G$) was 0.97, suggesting the graphitization of the Mn@CCs (Fig. 1d).

Then, we determined the molecular structure of Mn@CCs. The thermal gravimetric analysis (TGA) carried out in the nitrogen atmosphere further confirmed that about 8.3% of the mass was lost below 200 °C, which was attributed to the adsorbed moisture, and 50.7% of the mass was lost between 200 and 800 °C, corresponding to the decomposition of carbonaceous components (Fig. 1e). To understand the surface composition and electronic states of the constituent elements present in Mn@CCs, X-ray photoelectron spectroscopy (XPS) analysis was carried out. The survey XPS spectrum displayed four typical peaks at 284.8 eV (C1s, 72.2 atom %), 531.7 eV (O1s, 21.1 atom%), 399.7 eV (N1s, 5.2 atom%), and 641.2 eV (Mn2p, 1.5 atom%) (Fig. 1f). In the high-resolution XPS spectrum of Mn2p, the Mn2p peak was decomposed into four peaks at 641.0, 642.6, 652.8, and 653.9 eV (Fig. 1g). The resolved $Mn2p_{3/2}$ and $Mn2p_{1/2}$ values corresponded to the Mn-O bonds, which is consistent with the XPS of manganese gluconate (Supplementary Fig. 2). The high-resolution N1s XPS spectrum disclosed three different types of nitrogen atoms in Mn@CCs: pyridinic nitrogen (398.6 eV), secondary amine (399.5 eV), and graphitic N (401.4 eV) (Fig. 1h). The high-resolution C1s XPS spectrum provided evidence of sp3/sp2 carbon (C–C/C=C) and oxygenated carbon (C–O and C=O) (Fig. 1i). Two O1s bands corresponding to C–O (532.8 eV) and C=O (531.2 eV) were also identified (Fig. 1j). These results

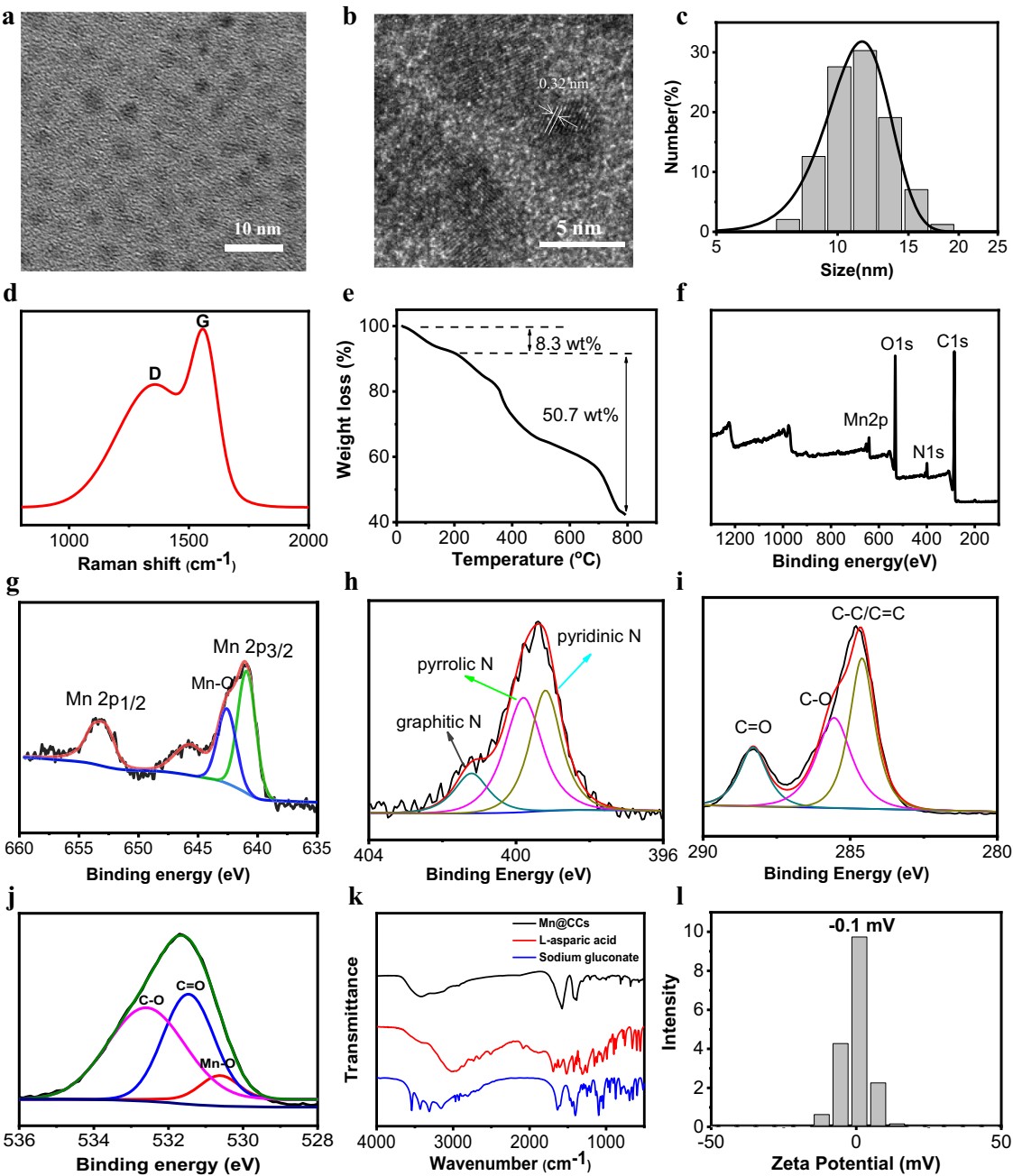

**Fig. 1 Characterizations of Mn@CCs. a** TEM image of Mn@CCs. **b** The graphene-like structure of Mn@CCs revealed by HR-TEM imaging. Experiments were repeated three times. **c** Size distribution of Mn@CCs. **d** Raman spectrum of Mn@CCs. **e** TGA of Mn@CCs. **f** The full-scan XPS spectrum of Mn@CCs. **g–j** High-resolution XPS spectra of **g** Mn2p, **h** N1s, **i** C1s, and **j** O1s. **k** FTIR spectra of Mn@CCs and raw materials. **l** Surface zeta potential of Mn@CCs.

confirmed that Mn@CCs not only successfully encapsulated and stabilized Mn ions but also had surfaces rich in oxygen and nitrogen, which is useful for further modification. In addition, the content of Mn in CDs, as determined by inductively coupled plasma mass spectrometry (ICP-MS), was about 5.9 wt%, which agreed well with the XPS results (Mn, 6.1 wt%). Fourier transform infrared spectroscopy (FTIR) found bands at 1560, 1388, and 1307 cm$^{-1}$, which were attributable to the stretching vibrations of C=O, C=C, and C–N groups, respectively (Fig. 1k). Meanwhile, the bands in the range of 3300–3500 cm$^{-1}$ and a relatively small peak at 2930 cm$^{-1}$ suggested the presence of N–H and O–H groups (Fig. 1k). These made the zeta potential of Mn@CCs close to neutral charge in an

aqueous solution, which indicates the presence of an approximately equal number of carboxyl and amino groups on their surface (Fig. 1l). After being stored at 4 °C for a whole year, it is still a clear and transparent solution (Supplementary Fig. 3). A near-neutral electric charge gave Mn@CCs good colloidal stability in different media (PBS [pH 7.4], Dulbecco's Modified Eagle's Medium (DMEM), and fetal bovine serum [FBS]) without any notable aggregation or precipitation after 3–7 days (Supplementary Fig. 4). The surface structures of Mn@CCs effectively minimize adsorption of serum protein and blood cells (Supplementary Figs. 5, 6), indicating there is no interference from serum protein and blood cells during circulation, and good blood compatibility[33,34].

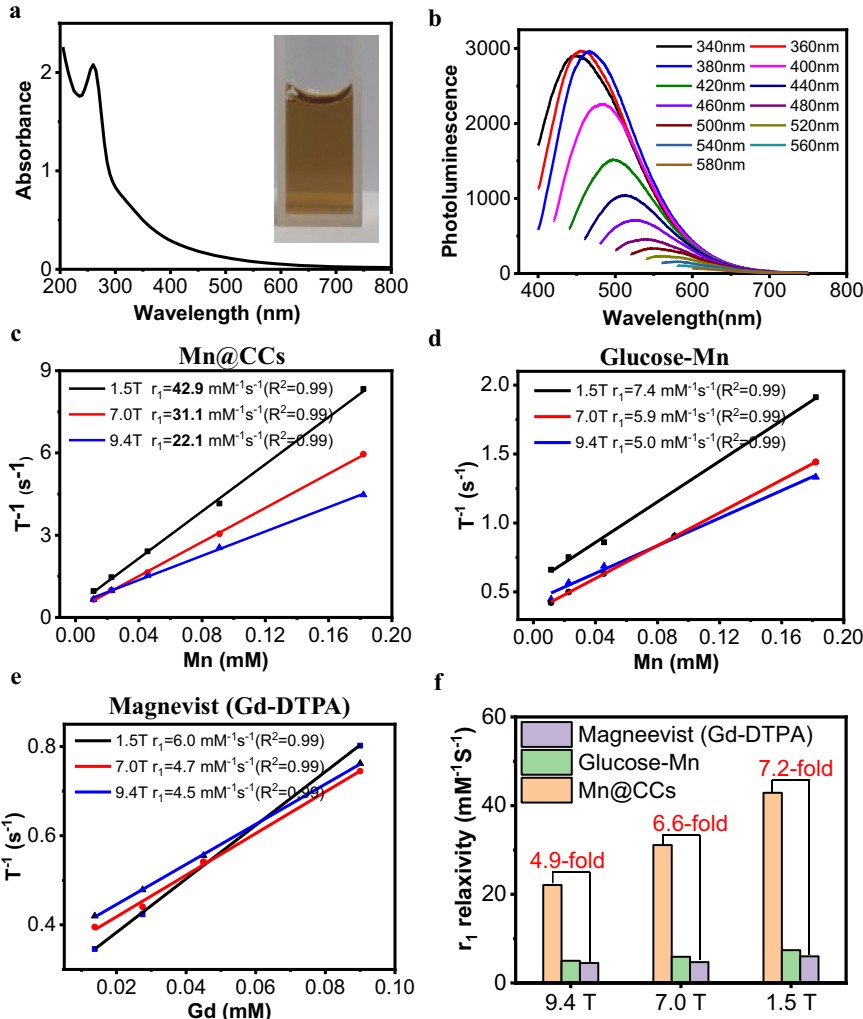

**Fig. 2 Optical and magnetic properties of Mn@CCs. a** UV-vis absorbance of Mn@CCs. Inset: Photograph of the purified Mn@CCs under daylight. **b** Photoluminescent spectra of Mn@CCs. **c** The $r_1$ relaxivities of Mn@CCs investigated in a 1.5 T, 7.0 T, and 9.4 T MRI system. **d** The $r_1$ relaxivities of glucose-Mn investigated in a 1.5 T, 7.0 T, and 9.4 T MRI system. **e** The $r_1$ relaxivities of Magnevist (Gd-DTPA) investigated in a 1.5 T, 7.0 T, and 9.4 T MRI system. **f** Comparison of $r_1$ relaxivities of Mn@CCs, glucose-Mn, and Gd-DTPA.

**Optical and magnetic properties of Mn@CCs.** The ultraviolet-visible absorption spectrum of Mn@CCs exhibited a strong characteristic absorption band centered at 280 nm, representing the n-π* transitions of the aromatic $sp^2$ system containing C=O and C=N bonds (Fig. 2a). Photoluminescence showed excitation-dependent luminescence at 400–650 nm (Fig. 2b). As the Mn content was as high as 5.9 wt%, MR phantom studies were conducted in 1.5 T, 7.0 T, and 9.4 T MRI scanners. The relaxations $(1/T_1)$ exhibited linear relationships with concentrations of Mn@CCs (Fig. 2c, Supplementary Fig. 7a), and the longitudinal relaxivities $(r_1)$ of Mn@CCs were up to 42.9 mM$^{-1}$s$^{-1}$ (1.5 T), 31.1 mM$^{-1}$s$^{-1}$ (7.0 T), and 22.1 mM$^{-1}$s$^{-1}$ (9.4 T) (Fig. 2c). The $r_1$ values were far higher than the precursor and the most common clinical MRI contrast agent Magnevist, i.e., Gd-DTPA (Fig. 2d, e, Supplementary Fig. 7). For instance, the $r_1$ value of Mn@CCs was 4.4 times higher than the precursor glucose-Mn and 4.9-fold higher than Magnevist under the 9.4 T MRI scanner (Fig. 2f), and even higher $r_1$ values were detected under lower magnetic fields (Fig. 2f).

**Gliomas and hepatocyte carcinoma cell lines take up Mn@CCs.** The feasibility of Mn@CCs in bioimaging applications was investigated. During incubation with human glioblastoma astrocytoma

and human Caucasian hepatocyte carcinoma cell lines (U87MG and HepG2) for 4 h, the Mn@CCs were taken up by cells and the cell-internalized Mn@CCs showed strong blue, green, and red emission under excitation at 405, 473, and 559 nm, respectively (Fig. 3a). Then the cells were dissociated and collected for MR phantom, which showed a 3.2-fold enhancement in the Mn@CCs incubation group versus 1.3-fold enhancement in the Gd-DTPA and 2.6-fold enhancement in the glucose-Mn incubation group compared to the vehicle control (Fig. 3b, c). These results indicated that Mn@CCs are easily uptaken by cancer cells with excellent targeting ability. Then, a cellular real-time multiphoton laser scanning microscope was used to achieve a more comprehensive and intuitive observation of the Mn@CCs taken up by U87MG cells (Supplementary Fig. 8, Supplementary movie 1). The investigation of the internalization pathway of Mn@CCs by using flow cytometry measurement showed that the cellular uptake mainly follows the energy-dependent endocytosis and lipid raft-mediated endocytosis (Supplementary Fig. 9)[35,36].

After confirming the excellent cellular uptake, the cytotoxicity of Mn@CCs was evaluated using an MTT assay. Compare to the safest MRI contrast agent in the clinic, Gd-DTPA, similar cellular viability was observed at doses as high as 100 μg mL$^{-1}$ of Mn@CCs incubating with cancer cell lines (Supplementary Fig. 10). Most

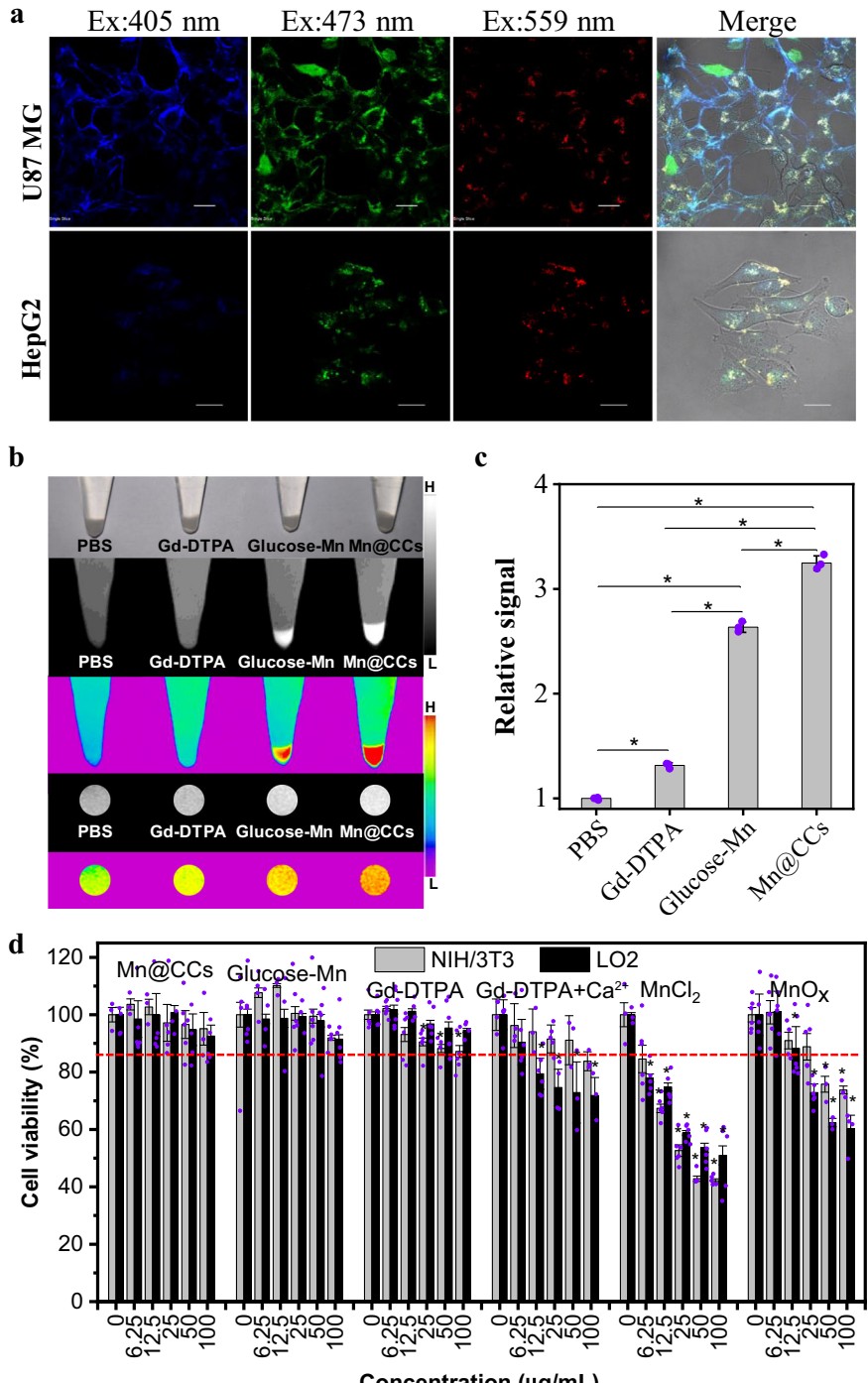

**Fig. 3 Gliomas and hepatocyte carcinoma cell lines take up Mn@CCs, and the toxicity assessments of Mn@CCs and other Mn-based agents.**
**a** Fluorescent images of U87 MG and HepG2 cell lines pretreated with Mn@CCs for 4 h at 37 °C under excitation of 405, 473, and 559 nm. Experiments were repeated three times. **b** Photograph and $T_1$-weighted MR images of U87MG cells incubated with PBS, Gd-DTPA, Glucose-Mn and Mn@CCs for 3 h. **c** Relative signal quantification of U87MG cells incubated with different agents in **b** ($n = 3$ independent experiments). *Significant differences from the PBS, Gd-DTPA, Glucose-Mn, and Mn@CCs with four group, $P < 0.05$. **d** Cell viabilities evaluated of normal cell lines (NIH/3T3 and LO2) incubated with Mn@CCs, Gd-DTPA, and Mn-based agents for 24 h. Experiments were repeated at least three times indicated as dot plots. *Significant differences from the control, $P < 0.05$. All the statistical data are expressed as mean values ± SD. Statistical significance was assessed via a one-way ANOVA with Duncan post-hoc test.

importantly, compare with Gd-DTPA, Mn@CCs showed minor cytotoxicity on normal cells, such as NIH/3T3 mouse embryonic fibroblast cell and LO2 human normal liver cell (Fig. 3d). Mn-based agents, including Mn salts and MnOx nanoparticles, showed significant cytotoxicity under the same conditions due to the generation of toxic reactive oxygen species (ROS) from the Mn-mediated Fenton-like mechanism (Fig. 3d)[18–20]. The above results and red blood cell hemolysis assay (Supplementary Fig. 6) further indicated Mn@CCs have high biocompatibility with no leaking of $Mn^{2+}$ ions into the medium to induce cytotoxicity[33,34].

**Mn@CCs cross the intact BBB in normal health mice**. Brain tumors are malignant tumors with a high incidence of nervous system diseases. The particularity of their location makes brain tumors highly disability-inducing and often fatal, and surgical treatment is currently the preferred method for intracranial tumors. However, the success of surgery is closely related to the determination of tumor biological boundary and its relationship with surrounding structures, so the precise diagnosis and differentiation of brain tumors directly affect the treatment outcome[37]. MRI technology is one of the main non-invasive anatomical or functional diagnostics for studying the development and diagnosis of tumorigenesis in the clinic because of its unlimited tissue penetration depth and greater soft-tissue contrast[38]. However, the BBB prevents 98% of small molecules and all large molecules from entering the brain, including currently available MRI contrast agents, such as Magnevist (Gd-DTPA)[30,31,39]. Although it is used in clinics for the detection of brain tumors, it is only used to detect tumor masses that have significantly damaged the BBB[39]. In previous studies, the surface analysis indicated that large carboxyl and amino groups were present on Mn@CCs, similar to the structure of an amino acid. Notably, mimicking biostructures on graphene-like dots gave the dots the ability to efficiently penetrate the BBB in vivo and achieve diagnostic and/or therapeutic roles for brain diseases[40–43].

Encouraged by cutting-edge research, we investigated whether or not our synthesized Mn@CCs could efficiently penetrate the BBB in normal healthy mice using MR imaging and multiphoton imaging. For MR imaging investigation, Mn@CCs or Magnevist with the dose of 20 mg·kg$^{-1}$ were injected by tail vain on normal mice ($n = 3$ biologically independent animals). Three regions of interest (ROIs) of the brain parenchyma (blue squares) and pituitary gland (red ellipsoid) in each $T_1$-weighted MR image were chosen to quantitatively evaluate the BBB-crossing ability of Mn@CCs (Fig. 4a)[30]. The signal enhancement of the pituitary gland in a 1 h period makes certain that both Mn@CCs and Magnevist enter the bloodstream (Fig. 4b, c). Since the pituitary gland does not have a BBB, the signal enhancement indicated that the Mn@CCs are capable of inducing similar MRI contrast as Magnevist[30]. Notably, the signal in brain parenchyma increased sharply with post-injection of Mn@CCs, however, compare with the injection of Magnevist, the signal from brain parenchyma remained almost unchanged as the Magnevist was blocked by the intact BBB (Fig. 4b, c). Even blue staining further confirms the integrities of BBB in these normal mice (Fig. 4d). All these results provided direct in vivo evidence that Mn@CCs penetrated an intact BBB and entered the brain parenchyma of living healthy mice[30].

To render the diffusion that Mn@CCs cross the intact BBB, we then performed multiphoton intravital live imaging through a cranial window (Fig. 4e) to assess the ability of Mn@CCs to cross the intact BBB in normal mice[44,45]. To minor visual artifacts induced from the cranial window (Fig. 4e), all image stacks were monitored at a typical depth of 120–164 µm from the surface. To monitor the vessel leakiness, we preinjected 70 kDa Texas red i.v. to assess vessel integrity following the cranial window procedure (red signals in Fig. 1f). Only red fluorescence seen in blood vessels and subsequent intravital images ensure vessel integrity. In contrast, the sharp green fluorescence of Mn@CCs was seen in a blood vessel and surrounding space (green spots in Fig. 1f), demonstrating Mn@CCs across the intace BBB into the surrounding brain space. Even blue staining further confirms the integrities of BBB in these normal mice (Fig. 4g). The histological section fluorescence image of brain tissue clearly showed the accumulation of Mn@CCs in the endothelial wall of a brain microvessel (purple square in Fig. 4h) with diffusion across the BBB and aggregation of Mn@CCs in the surrounding brain milieu (white arrows in Fig. 4h)[44]. The extravascular fluorescence was intensified along with the post-injection time of Mn@CCs (Supplementary Fig. 11a). The extravasation of Mn@CCs from vasculature could be clearly observed at 81 min post-injection, which indicated that Mn@CCs effectively crossed the BBB to reach the brain parenchyma. The mean fluorescence intensity in the selected regions increased from 21 min post-injection (Supplementary Fig. 11b). The whole process of probes crossing the intact BBB is more clearly shown in Supplementary Figs. 11, 12, and Supplementary movie 2.

**In vivo pharmacokinetics and biodistribution**. After confirming Mn@CCs' ability to penetrate the intact BBB in healthy mice, their biodistribution and clearance behavior were further evaluated by injecting Mn@CCs (20 mg kg$^{-1}$) in U87MG tumor-bearing mice and quantifying the fluorescence intensities in organs of interest. Fluorescence imaging was acquired at 0, 0.25, 0.5, 1, 2, and 4 h post-injection using an in vivo imaging system. Strong fluorescence signals were observed in the brain region; the intensity peaked at 0.5–2 h and then decreased gradually over time (Fig. 5a, b). Results ex vivo showed that significant portions of Mn@CCs were accumulated in tumors in the brain (Fig. 5c). The accumulation analysis of the main organs showed that almost all Mn@CCs have been extracted from the body after 24 h post-injection by detecting the amounts of Mn ions using ICP-MS (Fig. 5d, Supplementary Fig. 13). Additionally, the fluorescent signals in the kidneys increased and became intense (Fig. 5c, d), indicating the renal clearance of the Mn@CCs. Based on this finding, we detected fluorescence signals in the bladder and harvested the urine. Strong photoluminescence was found with the characteristic optical property of Mn@CCs (Fig. 5e, f). We also collected and photographed the urines, which showed clear brown color in the period 1–4 h, and gradually faded into normal yellow color. Both MR results and the analysis of Mn ions in the extracted urines showed similar trends (Supplementary Fig. 14). Overall, these results confirmed that Mn@CCs can efficiently pass through the BBB and accumulate in tumors in brain tissues, and then the unbound Mn@CCs are efficiently excreted through renal clearance, which is ideal for imaging purposes.

**Mn@CCs preferentially target the microscopic gliomas in an intracranial mouse model**. To evaluate the precise imaging capability of Mn@CCs for ultrasmall orthotopic tumors in mice, we established ultrasmall orthotopic brain tumors after 30-day inoculation and evaluated the tumors by bioluminescence and MRI (length: 2.7 ± 0.6 mm; width: 1.5 ± 0.3 mm, Supplementary Fig. 15). After tail vein injection of Magnevist or Mn@CCs (20 mg kg$^{-1}$), MRI signals in tumors were monitored and analyzed (Fig. 6a–j). Notably, clear increased intensity of the $T_1$-weighted MRI signal was observed in the whole brain, indicating that Mn@CCs successfully passed through the BBB (Fig. 6a, c, f, h). Most importantly, significant positive signal enhancement was observed in the tumor region in brain tissue with the size of 2.1 × 1.0 mm, which is successfully detected and clearly distinguished from the normal brain tissue up to 4 h post-injection of Mn@CCs (Fig. 6a, c, f, h). Consistent well with previous reports, Magnevist also shows a negligibly low uptake in the whole brain and gliomas (Fig. 6b, d, g, i). The main reason is that Magnevist are nonspecific contrast agents and only detect tumor masses that have significantly damaged the BBB[30,46,47]. Compare to Gd-DTPA, Mn@CCs showed high sensitivity and great accuracy to distinguish tumors and normal tissues (Fig. 6e, j). Moreover, the tumor size as determined by using Mn@CCs enhanced MRI is nearly identical to examine by the whole-brain hematoxylin and eosin (H&E) staining (Fig. 6a, f, k). The ultrasmall glioma of

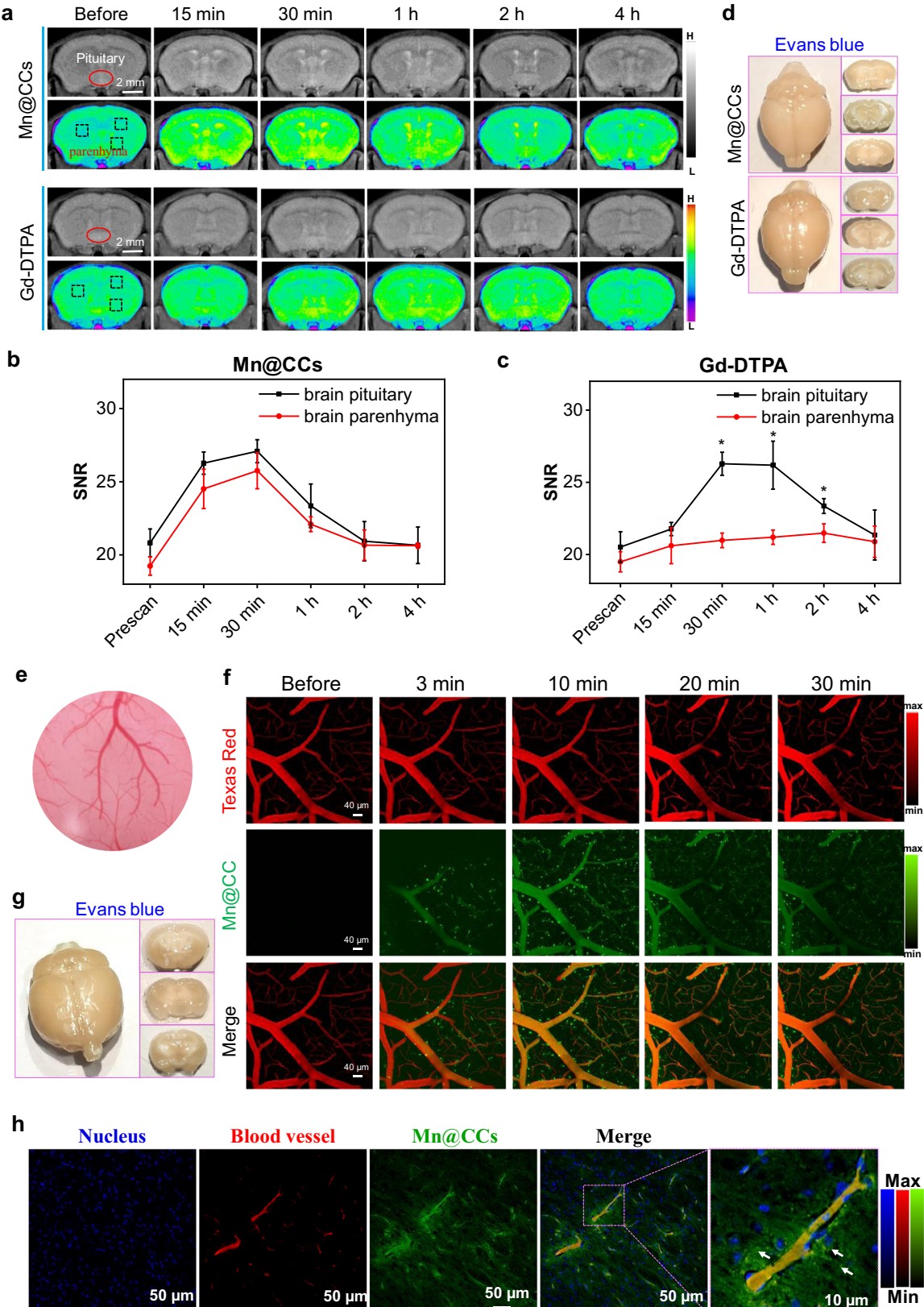

around 1 mm was clearly seen as well as distinguished from normal and benign tissue (Fig. 6k). An amplified image and detailed drawing of glioma cells confirmed the presence of the tumors (Fig. 6k, area 1, 2). These direct comparison results demonstrated that Mn@CCs provided superior tumor contrast enhancement to Magnevist (Gd-DTPA) in the ultrasmall orthotopic glioma mouse model, which achieved the accurate delineation of tumor region and its size by Mn@CCs enhanced MRI.

Dropping to pre-injection levels indicated the excretion of Mn@CCs from the circulation, and the signal duration was longer than that of Gd-DTPA[48]. Fluorescence imaging in vivo confirmed

**Fig. 4 Mn@CCs cross the intact BBB in normal health mice. a** In vivo MR images of normal mice at different timepoints after intravenous administration of Mn@CCs and Gd-DTPA. **b** Quantification of Mn@CCs MRI signals in brain parenchyma and pituitary gland of normal mice in **a** ($n = 3$ biologically independent animals). All the statistical data are expressed as mean values ± SD. No significant difference in Mn@CCs group. Statistical significance was assessed via an unpaired two-tailed $t$-test. **c** Quantification of Gd-DTPA in brain parenchyma and pituitary gland of normal mice in **a**. All the statistical data are expressed as mean values ± SD. *Significant difference compared with the brain parenchyma and pituitary gland: $P = 0.001$, 0.008, and 0.0170 at 0.5, 1 and 2 h, respectively. Statistical significance was assessed via unpaired two-tailed $t$-test. **d** The tracer, Evans blue, cannot permeates into the brain parenchyma in normal mice. **e** Photograph of cranial window exposing the brain for in vivo multiphoton imaging. **f** Intravital multiphoton imaging of brain of normal mice, showing diffusion of Mn@CCs (green signal) crossing BBB on normal mice. Texas red (red signal) were preinjected to label vasculature. **g** The tracer, Evans blue, cannot permeates into the brain parenchyma in normal mice. **h** Histology of brain tissues on normal mice, confirming Mn@CCs cross the intact BBB in normal mice ($n = 3$ biologically independent animals). Blue, nucleus; red, blood vessels; green, Mn@CCs.

that the Mn@CCs could be excreted from the renal system, so the MR signal changes in the bladder were also investigated. Results showed strong MR signals in the bladder, with the highest intensity at 2 h post-injection, indicating efficient renal clearance of the Mn@CCs (Fig. 6l, m). These results indicated that Mn@CCs have similar extraction efficacy as Magnevist with high safety.

The high imaging performance of Mn@CCs prompted us to further investigate their application for in vivo microscopic tumors diagnosis. Different from the above tumor sizes that were beginning to monitor at Day 30 post-inoculation of tumor cells, we established orthotopic gliomas with different tumor sizes (i.e., tumor volume) to simulate the early-stage gliomas after inoculating red fluorescence protein (RFP)-expressing U87 MG (U87 MG-RFP) cell lines at Day 5, Day 10, and Day 20 post-injection. After the injection of Mn@CCs, the MR images demonstrated clear visualization of the tumors of different sizes, even after only 5-Day inoculation (red arrows, Fig. 7a); Notably, multinodules existed in brain tissues were also clearly detected (Day 30, Fig. 7a). To show the morphology and structure of tumors, three-dimensional (3D) rendering of magnetic resonance images with the tumor segmentation was performed (orthotopic single-nodule gliomas indicated by green color and brain tissues indicated by gray color, Fig. 7b)[37]. After data processing and region segmentation accuracy measures, the tumor volumes were calculated, even the tumor cells were only inoculated on Day 5 (Fig. 7c)[49].

**Light-sheet fluorescence microscopy (LSFM) imaging allows 3D visualization of the distribution and penetration of Mn@CCs to tumors.** To clearly render the targeting, distribution, and penetration to tumors with different stages, the clearing and imaging technique and high-resolution mapping of the whole brain for the 3D reconstruction were performed, which enable complete observation of brain tumors[50,51]. After optical clearing using CUBIC[52], brain tissues are transparent, and tissue structures were visualized by light-sheet fluorescence microscopy (LSFM) imaging that performs wide-field 3D imaging of the entire tumor-bearing brain (Fig. 7d, e). In the 3D LSFM imaging, the red signal from U87 MG-RFP clearly labeled the tumors' position (Fig. 7e, Supplementary Fig. 16), and the supporting movie renders the tumors' morphology, clearing seeing the 3D structure at different stages (Supplementary video 3). Surprisingly, the green signal from our synthesized Mn@CCs overlapped very well with red signals from different angles (Fig. 7e, Supplementary Fig. 16). Visualization of the tumors in three dimensions clearly showed homogeneous distribution and high penetration in the whole tumors, even there are multinodules in late-stage of intracranial tumors (Supplementary movie 3). Collectively, these data show that Mn@CCs could be distributional across an intact BBB into the brain and achieve specific targeting, homogeneous distribution, and high penetration to intracranial tumors with different stages, which demonstrated that our Mn@CCs have a high potential to be considered in the development of effective diagnostics and therapies for GBM.

**Mn@CCs preferentially target the microscopic and dispersive hepatocellular carcinomas in an orthotopic mouse model.** Alongside orthotopic glioma, hepatocellular carcinoma is also one of the most common malignant tumors in the world. It has the highest degree of malignancy, ranking sixth in the incidence of malignant tumors and third in mortality rate, and thus seriously endangers life and health[53]. MRI is the best imaging method to detect the presence of a tumor capsule in the liver, and Gd-DTPA is also the most commonly used MRI contrast agent to achieve enhanced imaging[54]. Due to the excellent imaging capability of the Mn@CCs, we also established an ultrasmall and dispersive orthotopic hepatocellular carcinoma mouse model (total $6.5 \pm 1.4$ mm in length and $3.5 \pm 0.8$ mm in width) and monitored by bioluminescence and MRI (Supplementary Fig. 17).

To evaluate Mn@CCs' contrast enhancement performance and comparison with Gd-DTPA, we injected Magnevist or Mn@CCs ($20\,\mathrm{mg\,kg^{-1}}$) by tail vein, MRI signals in tumors were monitored, and analyzed (Fig. 8a–j). Remarkable positive signal enhancement was observed in orthotopic hepatic tissues at 1–4 h post-injection, suggesting that Mn@CCs successfully accumulated in liver tumors (Fig. 8a, c, f, h). Notably, more nodules smaller than 1 mm were easily detected and divided in the axial plane (enlarge images, bottom Fig. 8a). Then signals dropped to the pre-injection levels at about 8 h, indicating excretion of the particles from the circulation (Fig. 8a, c, f, h). It is also important to compare the efficacy of the Mn@CCs MRI contrast agent to the commercial gadolinium-based agent Gd-DTPA in the ultrasmall and dispersive orthotopic hepatocellular carcinoma mouse model. With the equal dose injection, only tumor enhancement was achieved with low SNR and quickly disappeared within 1 h post-injection (Fig. 8d, i). Also, the enlarged images could not delineate clear tumor margins (Fig. 8b, g). Compare to Gd-DTPA, Mn@CCs showed high sensitivity and great accuracy to distinguish tumors with multinodules and normal tissues (Fig. 8e, j).

The whole-liver H&E staining image showed that the microscopic and dispersive hepatocellular carcinoma with the size of around 0.5 mm was clearly seen as well as distinguished from normal and benign tissue, and an amplified image and detailed drawing of hepatic cells confirmed the presence of the tumors (Fig. 8k). These direct comparison results demonstrated that Mn@CCs also provided superior tumor contrast enhancement to Gd-DTPA in the microscopic and dispersive orthotopic hepatocellular carcinoma mouse model, which achieved the accurate delineation of tumor region, nodule numbers, and its size by Mn@CCs enhanced MRI.

**Long-term biocompatibility of Mn@CCs.** Furthermore, the long-term biocompatibility of Mn@CCs was evaluated by intravenous injection of Mn@CCs (60 mg/kg) into healthy mice. The histological examinations of the main organs, including the liver, spleen, kidney, heart, and lung, on day 7 using H&E staining showed no obvious disturbances (Supplementary Fig. 18a). Blood biochemistry and hematology analysis showed no significant

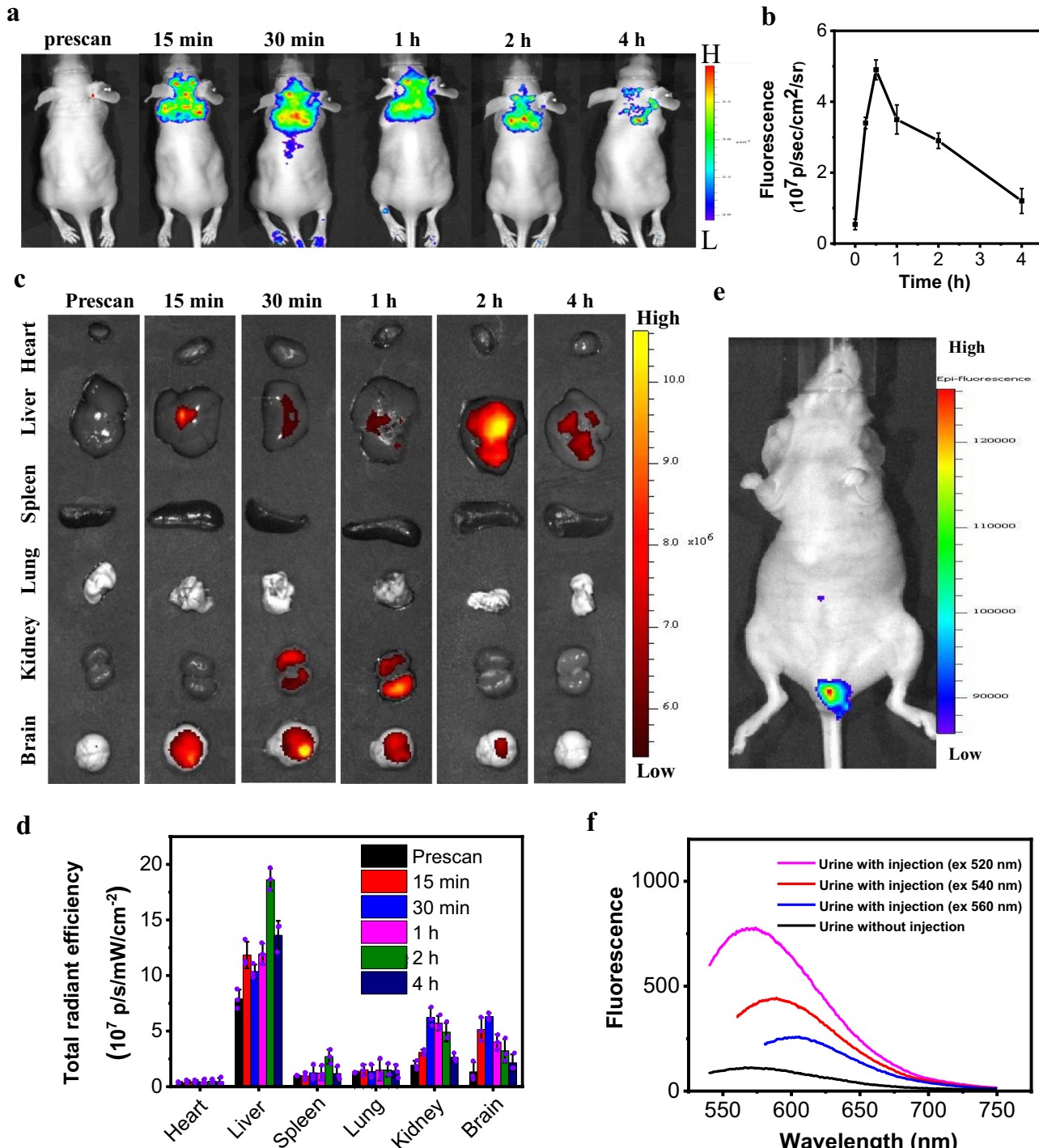

**Fig. 5 Evaluation of Mn@CCs preferentially target the microscopic gliomas in an intracranial mouse model by fluorescent imaging. a** The in vivo fluorescence images of U87MG glioma-bearing mice at different timepoints after intravenous administration of Mn@CCs (ex: 500 nm; em: filter dsRED). **b** The changes of the mean fluorescence intensity over time of Mn@CCs in brain parenchyma ($n = 3$ biologically independent animals). All the statistical data are expressed as mean values ± SD. The mean fluorescence intensity was calculated by ImageJ. **c** Ex vivo fluorescence images of organs excised at different timepoints post-i.v. injection of Mn@CCs. **d** Quantification of fluorescence signals of heart, liver, spleen, lung, kidney, and brain obtained from the ex vivo imaging data (ex: 500 nm; em: DsRed) ($n = 3$ biologically independent animals). All the statistical data are expressed as mean values ± SD. **e** Representative in vivo fluorescence image of the bladder area after i.v. injection of Mn@CCs at 1 h (ex: 500 nm; em: filter dsRED) ($n = 3$ biologically independent animals). **f** Fluorescent analysis of the urine samples obtained from the mice with and without injection.

difference in hepatic function, kidney function, or blood indexes before and after injection of Mn@CCs, indicating that Mn@CCs exerted no systematic toxicities (Supplementary Fig. 18b).

In summary, we produced carbonized paramagnetic complexes of manganese (II) (Mn@CCs) using food additives and an amino acid as the precursors. $Mn^{2+}$ ions were coordinated with functional groups to seal in carbonized shells, which exhibited superhigh $r_1$ relaxivity (22.1 $mM^{-1} S^{-1}$, 9.4 T) and high stability. Compared to the most common clinical contrast agent Magnevist (Gd-DTPA), we demonstrated that the Mn@CCs

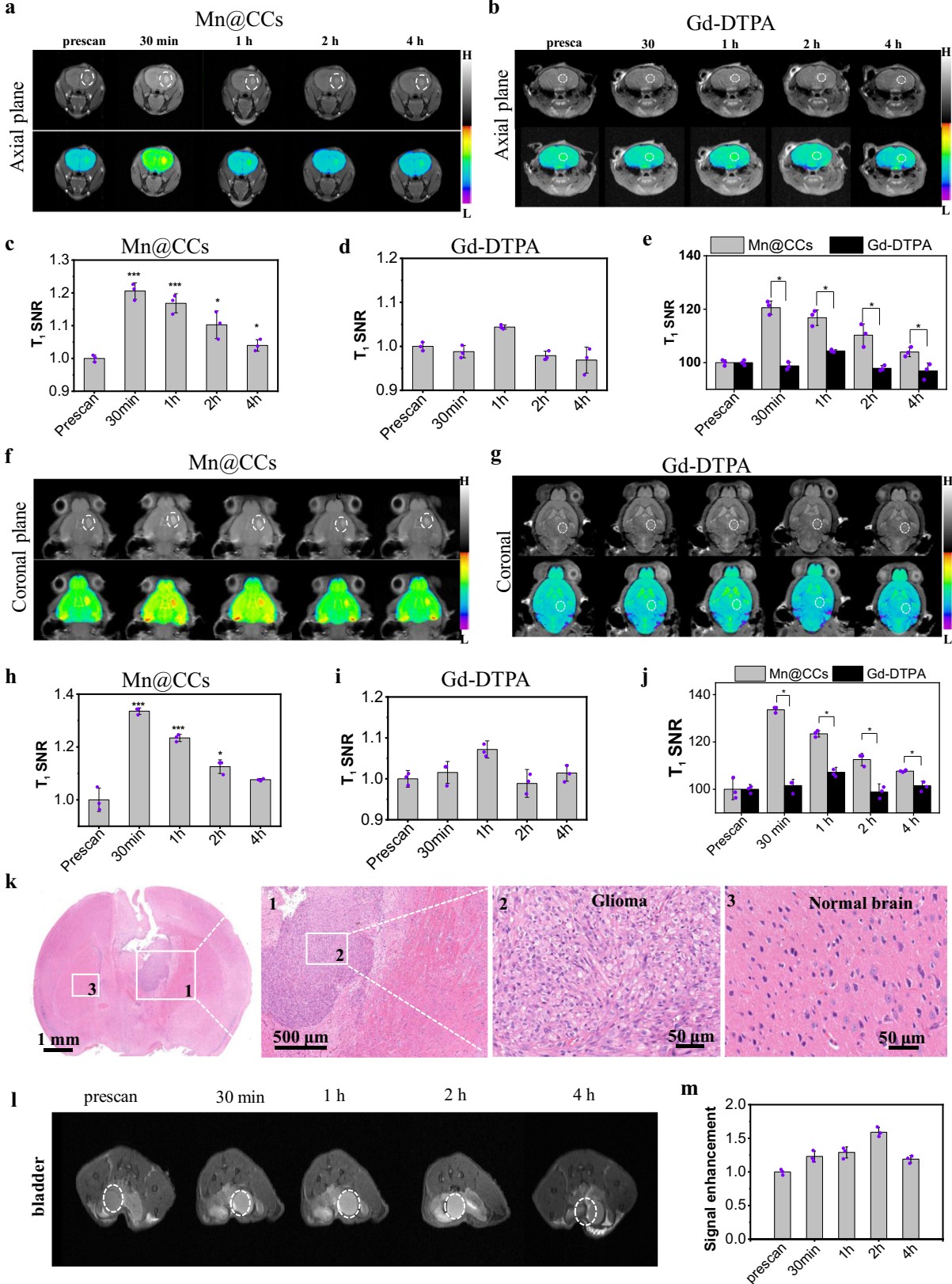

enable to cross the intact BBB in normal health mice; most importantly, 3D MRI and LSFM imaging allows three dimensions visualization of the homogeneous distribution and high penetration of Mn@CCs to tumors, demonstrating that Mn@CCs enable precise magnetic resonance imaging of microscopic orthotopic single-nodule brain tumors (about 1 mm) and multinodular liver tumors (each nodule < 1 mm) in mice. Moreover, Mn@CCs are efficiently excreted from the host at

**Fig. 6 Evaluation of Mn@CCs preferentially target the microscopic gliomas in an intracranial mouse model by MR imaging. a**, **b** In vivo axial MR images of U87MG glioma-bearing mice at different timepoints after intravenous administration of Mn@CCs and Gd-DTPA. **c**, **d** Quantification of MRI signals in glioma at different timepoints of Mn@CCs (**c**) and Gd-DTPA (**d**) in **a** and **b** ($n = 3$ biologically independent animals). All the statistical data are expressed as mean values ± SD. Significant difference compared with control. *$P < 0.05$, ***$P < 0.001$. Statistical significance was assessed via unpaired two-tailed Student's *t*-test. **e** Comparison of Mn@CCs and Gd-DTPA to image ability of microscopic glioma ($n = 3$ biologically independent animals). All the statistical data are expressed as mean values ± SD. *Significant difference compared with Mn@CCs and Gd-DTPA: 30 min ($p < 0.001$), 1 h ($p = 0.002$), 2 h ($p = 0.008$), 4 h ($p = 0.023$), respectively. *$P < 0.05$. Statistical significance was assessed via an unpaired two-tailed *t*-test. **f**, **g** In vivo coronal MR images of U87MG glioma-bearing mice at different timepoints after intravenous administration of Mn@CCs and Gd-DTPA. **h**, **i** Quantification of MRI signals in glioma at different timepoints of Mn@CCs (**h**) and Gd-DTPA (**i**) in **f** and **g** ($n = 3$ biologically independent animals). All the statistical data are expressed as mean values ± SD. Significant difference compared with control. *$P < 0.05$, ***$P < 0.001$. Statistical significance was assessed via unpaired two-tailed Student's *t*-test. **j** Comparison of Mn@CCs and Gd-DTPA to image ability of microscopic glioma ($n = 3$ biologically independent animals). All the statistical data are expressed as mean values ± SD. *Significant difference compared with Mn@CCs and Gd-DTPA: 30 min ($p < 0.001$), 1 h ($p < 0.001$), 2 h ($p = 0.003$), 4 h ($p = 0.042$), respectively. *$P < 0.05$. Statistical significance was assessed via an unpaired two-tailed *t*-test. **k** H&E staining of glioma tissue slides from nude mice. **l** In vivo MR images of bladder at different timepoints. **m** Relative signal change of bladder at different timepoints based on MR imaging results ($n = 3$ biologically independent animals). All the statistical data are expressed as mean values ± SD.

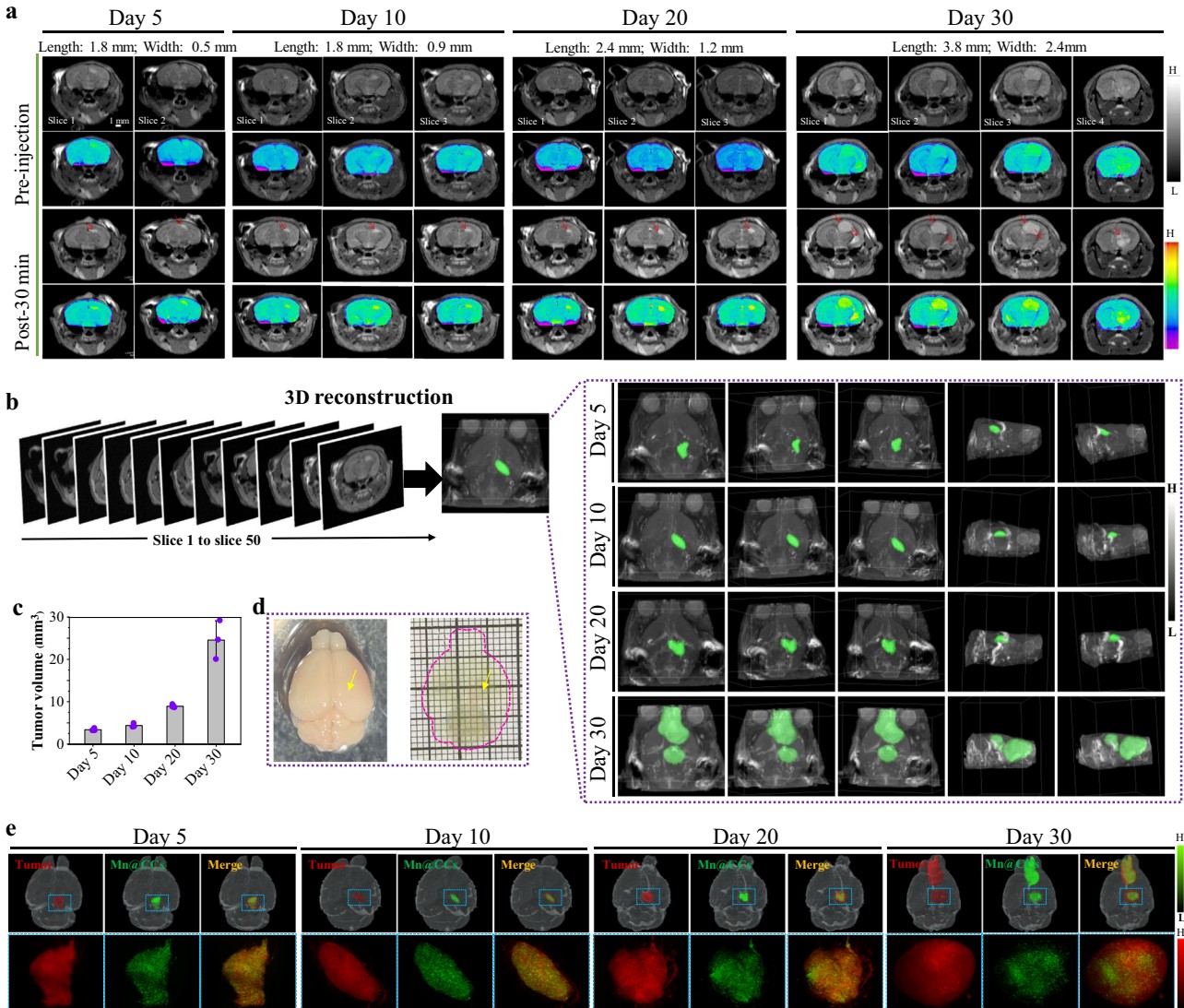

**Fig. 7 3D imaging allows visualization of the distribution and penetration of Mn@CCs to tumors. a** $T_1$-weighted MR imaging was performed before and after injection of Mn@CCs in different stages of orthotopic GBM tumor-bearing mice ($n = 3$). Day 5, Day 10, Day 20, and Day 30: The days that tumor cells were inoculated after 5, 10, 20, and 30 days to image. **b** 3D view constructed from Mn@CCs enhanced MRI of microscopic orthotopic gliomas (green color) in the brain tissues (gray color) under different angles. **c** Assessment of different early-stages different early-stages tumor volume by Amira 2020.1 (Thermo Fisher Scientific, USA) and Clinical Eclipse Treatment Planning System (15.6) ($n = 3$ biologically independent animals). All the statistical data are expressed as mean values ± SD. **d** Process of clearing and imaging brain and identification of small tumor (small arrows). **e** Wide-field image of whole brain from orthotopic GBM tumor-bearing mice. Gray: brain tissue, red: RFP-expressed glioma, Green: Mn@CCs.

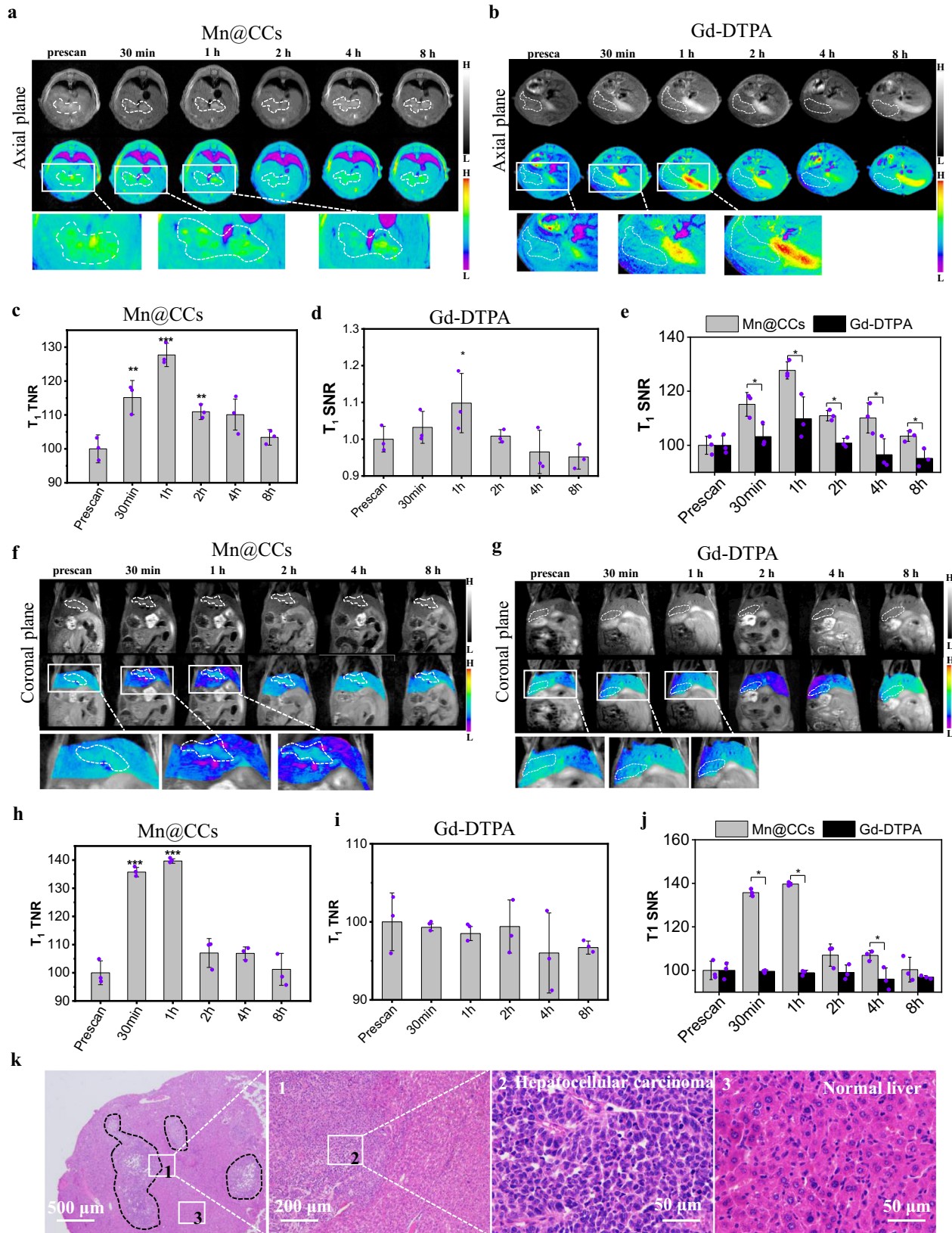

4 h post-injection through renal clearance with minor deposition. Collectively, the superior relaxivity and imaging ability of ultrasmall orthotopic brain and liver tumors could conclude that Mn@CCs are superior to the current clinical MRI contrast agents for orthotopic tumors and provide more reliable early-stage diagnostics.

## Methods

**Synthesis of Mn@CCs**. Manganese gluconate food additives (2 mmol) and L-aspartic acid (1.5 mmol) were dissolved in 20 mL of water and stirred for 30 min. Then the solution was transferred into a Teflon-lined stainless-steel autoclave and heated at 180 °C for 4 h. The suspension was centrifuged and the obtained dark brown supernatant was purified by dialysis (MWCO 1000). Then the dialysate was freeze-dried in the freezer, and finally the Mn@CCs were obtained.

**Fig. 8 Evaluation of Mn@CCs preferentially target the microscopic and dispersive hepatocellular carcinomas in an orthotopic mouse model. a, b** In vivo axial MR images of HepG2-bearing mice at different timepoints after intravenous administration of Mn@CCs (**a**) and Gd-DTPA (**b**). **c, d** Quantification of MRI signals in hepatoma at different timepoints of Mn@CCs (**c**) and Gd-DTPA (**d**) in **a** and **b** ($n = 3$ biologically independent animals). All the statistical data are expressed as mean values ± SD. Significant difference compared with control. $*P < 0.05$, $***P < 0.001$. Statistical significance was assessed via unpaired two-tailed Student's $t$-test. **e** Comparison of Mn@CCs and Gd-DTPA to image ability of microscopic hepatoma ($n = 3$ biologically independent animals). All the statistical data are expressed as mean values ± SD. *Significant difference Mn@CCs compared with Gd-DTPA: 30 min ($p = 0.029$), 1 h ($p = 0.023$), 2 h ($p = 0.003$), 4 h ($p = 0.044$), 8 h ($p = 0.021$), respectively. $*P < 0.05$. Statistical significance was assessed via an unpaired two-tailed $t$-test. **f, g** In vivo coronal MR images of HepG2-bearing mice at different timepoints after intravenous administration of Mn@CCs and Gd-DTPA. **h, i** Quantification of MRI signals in hepatoma at different timepoints of Mn@CCs (**h**) and Gd-DTPA (**i**) in **f** and **g** ($n = 3$ biologically independent animals). Significant difference compared with control. $*P < 0.05$, $***P < 0.001$. Statistical significance was assessed via unpaired two-tailed Student's $t$-test. **j** Comparison of Mn@CCs and Gd-DTPA to image ability of microscopic hepatoma ($n = 3$ biologically independent animals). *Significant difference compared with Mn@CCs and Gd-DTPA: 30 min ($p < 0.001$), 1 h ($p < 0.001$), 4 h ($p = 0.028$), respectively. $*P < 0.05$. Statistical significance was assessed via an unpaired two-tailed $t$-test. **k** H&E staining of orthotopic hepatic tissue slides from nude mice. Experiments were repeated three times.

**MRI phantom studies**. The Mn@CCs with various concentrations were suspended in 200 μL PCR tubes. The relaxivity and MRI phantom images at 1.5 T were performed on HT-MICNMR-60 system, which was acquired by a spin-echo (SE) sequence: TR (repetition time)/TE (echo time) = 100/8.3 ms ($T_1$), 128 × 128 matrices. The relaxivity and MRI phantom images at 7 T were recorded on a Bruker Biospin 7 T animal MRI scanner (Bruker, Germany), which were acquired under the following parameters: TR/TE = 300/10 ms ($T_1$), 128 × 256 matrices. The relaxivity and MRI phantom images at 9.4 T were recorded on a Bruker Biospin 9.4 T animal MRI scanner (Bruker, Germany), which were acquired under the following parameters: TR/TE = 300/10 ms ($T_1$), 128 × 256 matrices.

**Stability test of the Mn@CCs in the physiological environment**. To test whether these Mn@CCs bind to serum protein or not, the Mn@CCs were incubated with either phosphate-buffered saline (PBS) or PBS supplemented with 10% (v/v) fetal bovine serum (FBS) at 37 °C for 30 min. In order to identify the colorless protein band, FBS incubated-Mn@CCs and pure FBS were stained by 10% (v/v) Coomassie Brilliant Blue 250 (CBB 250). All these samples were analyzed using 10% polyacrylamide electrophoresis.

**Hemolysis test**. RBCs were obtained by centrifugation of freshly collected male mouse blood. The collected blood was resuspended in physiological saline. The Mn@CCs were added into the suspension at 37 °C for 2 h. Distilled water and physiological saline, in place of Mn@CCs, were used as a positive and negative control, respectively. After incubation, all blood samples were centrifuged, the supernatant was collected and the absorbance at 540 nm was measured. Hemolysis percentage was calculated according to the following equation:

$$\text{Hemolysis}(\%) = \left[\frac{\text{Ab}_{\text{sample}} - \text{Ab}_{\text{negativecontrol}}}{\text{Ab}_{\text{positvivecontrol}} - \text{Ab}_{\text{negativecontrol}}}\right] \times 100\%$$

**Blood-cell-binding test**. The whole blood was collected from male BALB/c nude mice (6–8 weeks, 20–25 g) and then stored in an anti-coagulation BD Vacutainer. This blood was divided into 5 × 3 groups for incubation with Mn@CCs at a 10% v/v ratio at room temperature for 30 min, followed by centrifugation at room temperature at 500 × $g$ to separate the plasma and blood cells. Finally, the amounts of $Mn^{2+}$ in plasma and blood cells were quantified by using ICP-MS.

**In vitro cellular imaging**. For fluorescent imaging, U87MG cells and HepG2 cells were plated on 10 mm glass-bottom dishes with an initial density of 2 × 10⁵ cells per mL and allowed to adhere for 12 h. Then, the culture medium was replaced by Mn@CCs (50 μg·mL⁻¹) for another 4 h incubation at 37 °C. The fluorescence images were acquired after washing cells using a laser scanning confocal microscope (Leica Microsystems, Germany) under the excitation light of 405, 473, 559 nm and a bright field. For MR imaging, 2 × 10⁵ U87-MG cells were incubated with 100 μg·mL⁻¹ Mn@CCs and 100 μg·mL⁻¹ manganese gluconate solution for 4 h at 37 °C, after incubation, the cells were washed with PBS for three times, and were precipitated at the bottom of the tube after centrifugation. The MRI experiments were performed on a 9.4 T MRI scanner.

For MR imaging, 2 × 10⁵ U87 MG cells were incubated with PBS, Gd-DTPA, Glucose-Mn, and Mn@CCs with the concentration of 100 μg·mL⁻¹ for 3 h at 37 °C, respectively. After incubation, the cells were washed with PBS for three times to remove free agents, and then were precipitated at the bottoms of tubes after centrifugation. $T_1$-weighted MR images were performed on a 9.4 T MRI scanner (Bruke 9.4 T MicroMRI) with parameter: echo time = 8.5 ms, effective TE = 8.5 ms, number of experiments = 8, repetition time = 1000 ms, flip angle = 180, rare factor = 4, number of repetitions = 1, number of averages = 6, matrix = 256 × 256.

**Cell lines**. U87MG glioma cell line, HepG2 hepatocellular carcinoma cell line, NIH/3T3, and LO2 cell lines were purchased from Sigma–Aldrich (USA). U87MG-RFP cell line was purchased from Shanghai Zhong Qiao Xin Zhou Biotechnology Co. Ltd.

**Cell cytotoxicity**. U87MG glioma cell line and HepG2 hepatocellular carcinoma cell line were seeded in a 96-well plate at a density of 1 × 10⁴ cells/well and allowed to adhere overnight, the media was then replaced by Mn@CCs with different concentrations (0, 6.25, 12.5, 25, 50, and 100 μg·mL⁻¹), the incubation was continued for another 24 h. Then, the cell viability was studied using the standard MTT assays performed by monitoring the absorbance of each well at 490 nm.

**Cell cytotoxicities of contrast agents on normal cell lines**. NIH/3T3 cell lines and LO2 cell lines were seeded in a 96-well plate at a density of 1 × 10⁴ cells/well and allowed to adhere overnight, the media was then replaced by Glucose-Mn, Mn@CCs, MnCl₂, MnOx nanoparticles, Gd-DTPA or Gd-DTPA (+2.5 mM $Ca^{2+}$) with different concentrations (0, 6.25, 12.5, 25, 50, and 100 μg mL⁻¹), the incubation was continued for another 24 h. Then, the cell viabilities were evaluated using the standard MTT assays performed by monitoring the absorbance of each well at 490 nm.

**Cellular internalization of Mn@CCs**. U87 MG cells were seeded in a 24-well plate at a density of 5 × 10⁴ cells/well. Cells untreated with Mn@CCs were denoted as the negative control group, while cells incubated only with Mn@CCs (50 μg mL⁻¹) at 37 °C were denoted as the positive control group. To study energy-dependent processes, the cells were first preincubated at 4 °C instead of 37 °C for 1 h. After that, nanoparticles were added at a concentration of 50 μg mL⁻¹. The mixtures were incubated for another 4 h at 4 °C. Finally, the cells were trypsinized, washed, and resuspended in 0.5 mL PBS in tubes for flow cytometer (FCM) analysis. To analyze the different endocytotic uptake mechanisms, the cells were first preincubated with inhibitions in serum-free media for 1 h at 37 °C: NaN₃ (60 mM), sucrose (450 mM), nystain (180 nM), and dynasore (80 μM). After that, Mn@CCs (50 μg mL⁻¹) were added and coincubated with inhibitors for 4 h at 37 °C. Finally, the cells were trypsinized, washed, and resuspended in 0.5 mL PBS in tubes for FCM analysis.

**Confirming study that $Mn^{2+}$ was encapsulated in carbonized shells, not deposited on the surface**. To confirm $Mn^{2+}$ was encapsulated in carbonized shells, the enzyme-like activity was measured using 3,3',5,5'-Tetramethylbenzidine (TMB), and $H_2O_2$ as substrates. Typically, acetic acid-sodium acetate buffer (pH 4.0, 0.2 M) containing 10 mM $H_2O_2$, 416 μM TMB, and 10 μg mL⁻¹ Mn@CCs were mixed thoroughly. After mixing, the reaction solution was immediately used for UV-vis spectroscopic measurements.

**Animal studies**. All animal experiments were carried out according to the criteria approved by Institutional Animal Care and Use Committee (IACUC) at Xiamen University. Male BALB/c nude mice were purchased from Shanghai SLAC Laboratory Animal Co. Ltd (Shanghai, China) at the age of around 5–6 weeks after birth and housed in 12 light/12 dark cycle, 65–75 °F (18–23 °C), 40–60% humidity condition. Tumor-bearing male mice were euthanized when the tumor reached 1.5 cm in diameter or when they became moribund with severe weight loss.

**Comparison of Gd-DTPA and Mn@CCs crossing the blood–brain barrier in normal mice using MR imaging**. MRI studies were carried out to evaluate the ability of Mn@CCs to penetrate an intact BBB in vivo with normal male mice. Two groups of animals ($n = 3$) were anesthetized with 2% isoflurane/oxygen and subsequently injected through the tail vein with either Magnevist (Gd-DTPA) or Mn@CCs (20 mg·kg⁻¹). Axial and coronal $T_1$-weighted MR images were acquired on a 9.4 T MRI system (Bruke 9.4 T MicroMRI) with echo time = 8.5 ms, effective

TE = 8.5 ms, number of experiments = 8, repetition time = 1000 ms, flip angle = 180, rare factor = 4, number of repetitions = 1, number of averages = 6, matrix = 256 × 256, FOV Read = 3 cm, slice thickness = 0.5 cm, slices = 50. The signal-to-noise ratios (SNRs), graphed as time course against SNR of pre-injection and post-injection images were evaluated from two regions of interest: (i) the posterior pituitary gland where the BBB is absent as an internal reference and (ii) the brain parenchyma on the same image slice. Evans blue staining was employed to assess the integrities of the blood–brain barrier in these normal mice.

**Assessment of Mn@CCs crossing the blood–brain barrier in normal mice using intravital multiphoton intravital imaging through a cranial window**. The BBB-crossing abilities of Mn@CCs were directly evaluated using intravital multiphoton imaging on normal mice. BALB/c nude mice (6–8 weeks old, male, $n = 8$) were anesthetized with 2–4% isoflurane. The skins of the skull were removed. Then, the cortex can also be visualized through open-skull windows after performing a craniotomy and replacement of the skull by a cover glass (4 mm diameter). Texas red (5 mg mL$^{-1}$, Thermo Fisher Scientific) was preinjected i.v. to outline the blood vessels, and Mn@CCs (20 mg/kg) were then injected and dripped with a drop of the ultrasonic coupler on the cover glass. Mice were fixed to a custom-built stage to minimize breathing artifacts during image acquisition. Multiphoton imaging was performed on a ×25 water immersion objective (Olympus FVMPE) on a multiphoton microscope equipped with a tunable pulsed chameleon infrared multiphoton laser (Coherent). High-resolution xyz-stack images (1024 × 1024 pixels per z-step) were taken with a step size of 2 μm, a z-stack was acquired 120 μm to a depth of 164 μm and XYZT stacks were taken every 3 min for cycled 10 times. A preprogrammed program was executed every 3 min, taking 23 images in succession once time. Acquired images were then imported into Imaris (Bitplane, Belfast UK) and Image J for further analysis. After imaging, Evans blue (2%, 2 ml/kg) was injected i.v. to stain the blood–brain barrier. To validate the in vivo results, histology of the brain on normal mice was performed. The blue color indicated the nucleus stained with DAPI, the red color indicated the blood vessels stained with Texas Red (70000), and the green color indicated the Mn@CCs.

**Orthotopic tumor models**. Orthotopic glioma was established: Male nude mice (BALB/c, 18–20 g) were anesthetized with 5% chloral hydrate and placed onto a stereotaxic apparatus (RWD, Life Science Co., Ltd). A burr hole was drilled into the skull (Bregma 2.0 mm, right lateral 2.0 mm, depth 2.5 mm), 10 μL of PBS containing $2 \times 10^6$ U87MG cells were slowly injected into the right side of the brain of the mice. The growth of glioma-bearing tumors was monitored by bioluminescence imaging and MRI. Typically, after 30-Day inoculation, the orthotopic glioma was successfully established and used in the following in vivo imaging.

Orthotopic hepatocellular carcinoma was established: Male nude mice (BALB/c, 18–20 g), were used for assessing all imaging. Briefly, mice were anesthetized with 5% chloral hydrate, and then 20 μL HepG2 cells ($1 \times 10^7$ per mouse) were injected into the right liver lobe by a laparotomy. The growth of HepG2-bearing brain tumors was monitored by bioluminescence imaging and MRI.

**In vivo imaging of microscopic orthotopic glioma by fluorescent and MR imaging**. Mn@CCs and Gd-DTPA (20 mg·kg$^{-1}$) were injected intravenously (i.v.) into U87MG glioma-bearing male mice ($n = 5$). For MR imaging, axial and coronal T$_1$-weighted MR images were acquired at different times post the nanoparticle-injection. The images were acquired using the following parameters: FOV (field of view) = 4.0 × 4.0 cm, TR/TE = 3000/10 ms, Matrix = 256 × 256, Average times = 1, Slice Thickness = 1 mm.

In the fluorescent biodistribution study, the same dose of the Mn@CCs were injected, fluorescent images at 0, 0.25, 0.5, 1, 2, and 4 h post-injection was observed through an in vivo imaging system (IVIS Lumina II, Caliper, USA; Ex: 500 nm, Em: DsRed). At 1 h, in vivo fluorescence image of the bladder area was acquired and the urine was also collected for ex vivo fluorescence analysis.

**Orthotopic red fluorescence protein (RFP)-expressing glioma models**. For the orthotopic intracranial red fluorescence protein (RFP)-expressing glioma model, 5–6-week-old male athymic nude mice were anesthetized with inhalational 1–5% isoflurane mixed with oxygen placed in a prone position in a stereotactic apparatus. RFP-expressing U87 MG (U87 MG-RFP) cell lines ($1 \times 10^6$ cells per mouse in 10 μl of cold PBS) were stereotactically injected into the right hemisphere of the mouse brain (1.8 mm lateral, 1 mm posterior to the bregma; 2 mm depth) using a 33-gauge needle attached to a stereotaxic frame.

**Three-dimensional (3D) MR imaging of whole brain of RFP-expressing glioma-bearing mice**. 3D MRI studies were carried out to evaluate the ability of Mn@CCs to target and detect orthotopic glioma. After U87 MG-RFP cells were injected into the mouse brain on the post-injection timepoints (term as Day 5, Day 10, Day 20, and Day 30), four groups of male mice ($n = 3$/group) were anesthetized with 2% isoflurane/oxygen and subsequently injected through the tail vein with either Mn@CCs (20 mg·kg$^{-1}$). Axial and coronal T$_1$-weighted MR images were acquired on a 9.4 T MRI system (Bruke 9.4 T MicroMRI) with echo time = 8.5 ms, effective TE = 8.5 ms, number of experiments = 8, repetition time = 1000 ms, flip angle = 180, rare factor = 4, number of repetitions = 1, number of averages = 6,

matrix = 256 × 256, FOV Read = 3 cm, slice thickness = 0.5 cm, slices = 50. 3D MR imaging reconstruction images and tumor volume calculation were generated using Amira 2020.1 (Thermo Fisher Scientific, USA).

**Tissue clearing protocol of whole brain of RFP-expressing glioma**. After MR imaging of whole brain of RFP-expressing glioma-bearing male mice, the mice were sacrificed. The heart was perfused with cold PBS and 4% paraformaldehyde (PFA) and brain tissue was extracted. The brain tissue was gently shaken overnight at 4 °C with 4% PFA. Then, the brain tissue was washed three times with PBS. Then use CUBIC reagent solutions 1, 2, 3 (Nuohai Life Science Co.) for tissue clearing. First, the brain tissue was gently shaken with CUBIC reagent solution 1 at 37 °C for 4 days. Then wash the brain tissue with PBS, and use solutions 2 and 3 in turn to continue to ease the shaking.

**3D fluorescent imaging of whole brain at single-cell resolution**. 3D fluorescent imaging of the above clearing tissue was acquired with Nuohai LS 18 light-sheet microscopy (Nuohai Life Science (Shanghai) Co., Ltd, laser lines: 405, 488, 561, 637 nm) with a ×1/0.25NA objective (Olympus MVPLAPO). At ×6.3 zoom effective magnification, each sample was scanned by 6-tiles light sheets axially at a 2.5 μm Z-step size. Fluorescent signals of Mn@CCs and tumors (U87 MG-RFP) were obtained by excitation at both 488 nm and 561 nm, respectively. All raw image data were collected in a dcimg format by LS 18 Nobelium software (v1.0.4) and those tiling of multiple position image stacks were processed into 16-bit 3D tiff format data using LS 18 Image Combine software (v1.0.1).

**Image processing and 3D reconstruction analysis**. Imaging processing and 3D rendering were performed with a Dell Precision 7820 workstation with Xeon Gold 5118 processor, 192 GB RAM, and NVIDIA Quadro P4000. 3D reconstruction images were generated using Amira 2020.1 (Thermo Fisher Scientific, USA). Maximum intensity projection of entire sample subvolumes 3d tiff images within the field of view was generated individually using the "Image Ortho Projections" module before stitching. The 3D images of all subvolumes were resampled down to ~3 × 3 × 3 μm voxel size resolution for gliomas tissues and ~5 × 5 × 5 μm for whole mouse brains before being registered and merged. The final 3D stack images were generated using the "voltex" module. Optical slices were obtained using the "Ortho Slice" module. Movies were generated using "MovieMaker" module with 25 fps.

**In vivo imaging of microscopic orthotopic hepatocellular carcinoma by MR imaging**. Mn@CCs and Gd-DTPA (20 mg·kg$^{-1}$) were injected intravenously (i.v.) into HepG2-bearing male mice ($n = 5$). For MR imaging, axial and coronal T$_1$-weighted MR images were acquired at different times post the nanoparticle-injection. The images were acquired using the following parameters: FOV (field of view) = 4.0 × 4.0 cm, TR/TE = 3000/10 ms, Matrix = 256 × 256, Average times = 1, Slice Thickness = 1 mm.

**Biodistribution and clearance behaviors in mice by fluorescent imaging**. Male BALB/c mice were intravenously injected with Mn@CCs (20 mg·kg$^{-1}$). At 0, 0.5, 1, 2, and 4 h post-injection, mice (n = 3/group) were euthanized and the major organs (heart, livers, spleen, lungs, kidneys, brain) were collected and we employed fluorescence imaging to investigate the biodistribution. Ex vivo fluorescence imaging of major organs was conducted and unmixed by the vendor-provided software. ROIs were circled around the organs and the optical intensities were read by the software. The urine was also collected to confirm the extraction of Mn@CCs.

**Biodistribution and clearance behaviors in male mice by ICP-MS**. In the elemental analysis group, the timepoints were extended to 8, 12, 24, and 48 h, and the weighed major organs were digested in freshly prepared aqua regia with heating until the solution became transparent. And the obtained solutions were diluted to 10 mL by adding ultrapure water. The samples were analyzed by inductively coupled plasma mass spectrometry (ICP-MS). The percentage of injected dose per gram of organ (%ID/g) was calculated by comparing the concentrations in samples with that of the ID.

For in vivo renal clearance kinetics study, male BALB/c mice were intravenously injected with 20 mg·kg$^{-1}$ of Mn@CCs (n = 3), respectively, and then placed in metabolism cages. Mouse urine was collected at different post-injection timepoints, which was measured by MRI and then completely lysed in freshly made aqua regia in a plastic centrifuge tube (15 mL) for 2 days. After diluting to 10 mL using ultrapure water, all of the above samples were analyzed by ICP-MS.

**In vivo safety evaluation**. Healthy male mice after i.v. injection of Mn@CCs (60 mg·kg$^{-1}$, three-fold dose) and saline were used as the experimental group and control group (n = 3), respectively. The major organs were harvested on day 7 post-injection, fixed in paraformaldehyde in PBS (4%), and stained with hematoxylin and eosin (H&E). The stained sections were observed under a microscope (Nikon Eclipse E600, Nikon Inc., Melville, NY) at ×100 magnification.

**Serum chemistry analysis**. Male BALB/c mice were intravenously injected with Mn@CCs (60 mg·kg⁻¹). Blood samples were collected for serum chemistry analysis on days 3 and 7 post-injection. The serum was collected by centrifugation at 8000 rpm for 10 min. Alkaline phosphatase (ALP), albumin (ALB), creatinine (CREA-S), and creatine kinase (CK-IFCC) were measured in serum using an Auto Biochemistry analyzer (Mindray, BS-220). White blood cell (WBC), red blood cell (RBC), hemoglobin (HGB), mean platelet volume (MPV), and large platelet ratio (P-LCR) were measured using an Automatic Tri-classification Blood Cell Analyzer (Mindray, BC-2600).

**Statistical analysis**. Data analysis was performed by Origin 9.1 or 2018 (64-bit) SR1-b.9.6.1.195, GraphPad Prism 6.0, SPSS statistics 22.0 software, and software Image J2x V2.1.4.7, NovoExpress for windows, Amira 2020.1 (Thermo Fisher Scientific, USA). Quantitative results were presented as mean values ± SD.

**Reporting summary**. Further information on research design is available in the Nature Research Reporting Summary linked to this article.

## Data availability

The imaging data are available from the corresponding author upon request. The authors declare that all other data supporting the findings of this study are available in the paper, its supplementary information files. Source data are provided with this paper.

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

## Acknowledgements

The work was supported by the National Science Foundation of China (82172007, 81771977, and 82001956), the Science Fund for Distinguished Young Scholars of Fujian Province (2021J06007), the National Postdoctoral Program for Innovative Talents (BX20200196), Xiamen Science and Technology Plan Project (3502Z20183017), and the Fundamental Research Funds for the Central Universities of China (20720180054), and the open research fund of National Facility for Translational Medicine (Shanghai) (TMSK-2021-102). All animal experiments were approved by the Animal Management and Ethics Committee of the Xiamen University.

## Author contributions

R.Q., S.L., Y.Q., and H.C. conceived the idea and supervised the research; R.Q., S.L., Y.Q., Y.F., D.D., L.X., X.M., and W.S. performed the synthesis and characterization; R.Q., S.L., and Y.Q. worked on orthotopic tumor models; R.Q., S.L., Y.Q., and Y.F. investigated the MRI, fluorescent imaging investigation; R.Q., S.L., Y.Q., Y.L., and H.C. analyzed the data; R.Q., S.L., Y.Q., and H.C. co-wrote the paper. All authors discussed the results and commented on the manuscript.

## Competing interests

The authors declare no competing interests.
