## [Peer Review File · Nature Communications]

Reviewers' comments:

Reviewer #1 (Remarks to the Author): Expert in MRI and contrast agents

This manuscript carbonized manganese complexes as new MRI contrast agents. While the results are somewhat interesting this manuscript a number major limitations preventing its publication in this journal.

1. Manganese-based MRI contrast agent are relatively new but are certainly not novel in the field of MRI. In fact several recent reports describe manganese-based MRI agents that appear to be quite similar to the agents proposed in this manuscript. (Tao Y, *Nanoscale* 2018; Dai C, *ACS Nano* 2017). Many of these prior publications have not been cited or differentiated from the current work. Therefore the novelty of this manuscript is greatly lessened.

2. Figure 2d,e: the noted r_1 values in the legend do not match the slopes of the lines in the plots.

3. Figure 3d shows that there is some drop in survival of the cells at higher concentrations. However, the manuscript says that there is no visible toxicity which seems inconsistent. Also, a comparison with Gd-DTPA and other Mn-based MRI agents may be warranted,

4. The manuscript claims that this agent gets past the BBB. The fluorescence results in Figure 4 seem to support this. But, then the same claim is made with the orthotopic tumor models where the BBB is already perturbed by the implanted tumor. In this case almost all MRI contrast agents will enhance the tumor.

5. The in vivo MRI images and data in Figures 5 and 6 are reasonable. But, these figures also do not demonstrate any new capability to differentiate from existing literature. All results seem fairly typical with existing technologies.

6. A spelling and grammer check throughout is needed.

Reviewer #2 (Remarks to the Author): Expert in brain imaging

This manuscript reports on an important advancement concerning positive contrast agents for magnetic resonance imaging (MRI). Authors synthesize and characterize carbonized complexes of Mn, showing superior performances over commercially available contrast agents.

Despite the importance of the topic and the interesting data, I think that the study, even if of high quality, does not show the potential impact necessary to be published on Nature Communications.

- 1) First of all, Authors should provide better evidences about stability of their nanoparticles: presenting a Z-potential around 0 mV, I am wondering for example about long-term stability, and interactions with serum / blood proteins
- 2) Again concerning stability, in methods Authors are describing enzyme-like activity. Please provide explanations about the meaning.
- 3) Biocompatibility is cursorily approached, and just a short-term MTT assay on cancer cell is provided. What about effects on healthy cells?
- 4) Nanoparticle/cell interaction investigations are cursory as well: uptake mechanisms and long-term fate analysis are completely missing.
- 5) A quantitative and long term evaluation of biodistribution is missing.
- 6) Statistical analysis is not adequate. At least a post-hoc correction for multiple comparison should be performed.
- 7) Language needs important improvements in many points.

Response to Reviewers

Dear Editor and Reviewers:

We highly appreciate the helpful and insightful comments on our manuscript entitled "*Carbonized paramagnetic complexes of Mn (II) as contrast agents for precise magnetic resonance imaging of sub-millimeter-sized orthotopic tumors*" (No. NCOMMS-21-10809). The revisions have been highlighted (yellow highlight) in the revised manuscript. The detailed point-by-point responses to the reviewers' comments are listed as follows.

Reviewer #1 (Remarks to the Author): Expert in MRI and contrast agents

This manuscript carbonized manganese complexes as new MRI contrast agents. While the results are somewhat interesting this manuscript a number major limitations preventing its publication in this journal.

1. Manganese-based MRI contrast agent are relatively new but are certainly not novel in the field of MRI. In fact several recent reports describe manganese-based MRI agents that appear to be quite similar to the agents proposed in this manuscript. (Tao Y, Nanoscale 2018; Dai C, ACS Nano 207). Many of these prior publications have not been cited or differentiated from the current work. Therefore the novelty of this manuscript is greatly lessened.

Our response:

Although it is uncertain of links between gadolinium contrast agents and patients reporting side effects, but there is growing cause for concern. As a result, there is an extra impetus to find better alternatives. One option is to use manganese (Mn) as it is an essential human dietary element, and also has essential roles in cell biology. As we know, Mn^{2+} possess the second-highest paramagnetic moment of any element, which make it an attractive alternative to Gd^{3+} in the design of contrast agents for medical MRI.

Though manganese-based magnetic resonance imaging (MRI) agents (mainly Mn oxides) have been prepared before, neither the wet-chemistry synthesis approaches nor the subsequent bioimaging by the decomposition of the oxide is fully explored.^[1] **The biocompatibility of manganese-based MRI agents is still a big challenge to achieve safe and efficient diagnosis via MR imaging.**^[2] As listed by Reviewer 1, in these studies, Mn oxides are degraded in tumor environment (acidic, high GSH and H_2O_2 level) to release Mn ions,^[3-5] which induce toxicity concerns as free Mn ions would generate toxic reactive oxygen species (ROS) via Fenton-like reaction.^[6-8] Safety concerns on MRI contrast agents made parts of metal-based complexes MRI contrast agents (mainly Gd-based complexes, iron oxides, $MnCl_2$) withdraw from the market or discontinued in medical use.^[9,10]

We cited and differentiated from the current work in page 4 line:

"Although strategies have been developed to enhance their sensitivity on their oxides (i.e. MnOx), its structures could be degraded into free Mn ions to achieve

significant enhancement of MRI, the significant toxicity from Mn ions via Fenton-like reaction have been widely reported and limited their translational applications.¹⁸⁻²⁰

In order to ensure in vivo safety and high contrast agent efficiency, Mn²⁺ ion has to be chelated by a ligand or cage (layer).^[10] Base on the above discussion, please allow us to highlight the innovative points of this article:

First, Mn can be toxic, so creating a stable platform for its delivery in the body is critical. In our manuscript, carbonized paramagnetic complexes of Mn (II) are constructed using food additives and an amino acid as the precursors, which has been rarely reported yet. The Mn (II) was encapsulated in sealed carbonized shells to form stable cages for Mn ions. Previous investigations have approved that these host-guest structures of gadofullerene showed higher stability of Gd ions and almost no toxicity in normal cells and body.^[11,12] Moreover, previous investigations by us and others confirmed that carbonized polymerized shells could encapsulate Gd ions to minimize leakage and achieve more better dispersibility in physical environment than gadofullerene as more surface hydrophilic functional groups are existed on its surface.^[13-15] So, **sealed carbonized shells enhance the stability of Mn ions.**

Second, so equally as important, the manner in which Mn sits within the carbonized polymerized shells provides a more optimal physical environment in which Mn can interact with a magnetic field, thereby boosting signal enhancement. As shown in Figure 2, the longitudinal relaxivities (r_1) of Mn@CCs were up to 42.9 mM⁻¹s⁻¹ (1.5 T), 31.1 mM⁻¹s⁻¹ (7.0 T), and 22.1 mM⁻¹s⁻¹ (9.4 T) (Figure 2c). The r_1 values were far higher than the precursor and the most common clinical MRI contrast agent Magnevist, i.e., Gd-DTPA (Figure 2f). This interesting enhanced property has been rarely reported.

Fig. 2c-f. (c) The r_1 relaxivities of Mn@CCs investigated in a 1.5 T, 7.0 T, and 9.4 T MRI system. (d) The r_1 relaxivities of glucose-Mn investigated in a 1.5 T, 7.0 T, and 9.4 T MRI system. (e) The r_1 relaxivities of Magnevist (Gd-DTPA) investigated in a 1.5 T, 7.0 T, and 9.4 T MRI system. (f) Comparison of r_1 relaxivities of Mn@CCs, glucose-Mn, and Gd-DTPA.

Moreover, our agent successfully crossed the intact (normal) blood–brain barrier (BBB) of health mice, confirmed by in vivo multiphoton imaging on health mice that cranial window exposes the brain (new Figures 4e-h, S11, S12). Normally, as reported, clinical Gd(III) chelates and >99% chemotherapeutic drugs do not pass the intact (normal) BBB, as the BBB is not really disrupted in all glioblastomas.^[16-18] As at the early stage of glioma, they cannot be enhanced by the MRI contrast agent in most cases because the BBB has not been significantly damaged, as indicated by a critical assessment of existing clinical data.

Our results successfully demonstrated that the Mn@CCs across an intact BBB into brain, achieving the critical first step and a key consideration in developing effective diagnostics and therapies for GBM.

Ref.: Jann N Sarkaria, Leland S Hu, Ian F Parney, Deanna H Pafundi, Debra H Brinkmann, Nadia N Laack, Caterina Giannini, Terence C Burns, Sani H Kizilbash, Janice K Laramy , Kristin R Swanson, Timothy J Kaufmann, Paul D Brown, Nathalie Y R Agar, Evanthia Galanis, Jan C Buckner, William F Elmquist. **Is the blood–brain barrier really disrupted in all glioblastomas? A critical assessment of existing clinical data**, Neuro Oncol. 2018, 20,184.

Fig. 4. Mn@CCs cross the intact BBB in normal health mice. (a) In vivo MR images of normal mice at different time points after intravenous administration of Mn@CCs and Gd-DTPA. (b) Quantification of Mn@CCs MRI signals in brain parenchyma and pituitary gland of normal mice in a (n=6). No significant difference in Mn@CCs group. Statistical significance was assessed via an unpaired two-tailed t-test. (c) Quantification of Gd-DTPA in brain parenchyma and pituitary gland of normal mice

*in a. *Significant difference compared with the brain parenchyma and pituitary gland: $P = 0.001$, 0.008 , 0.0010 , and 0.0170 at 0.5, 1 and 2 h, respectively. Statistical significance was assessed via unpaired two-tailed t-test. (d) The tracer, Evans blue, cannot permeates into the brain parenchyma in normal mice. (e) Photograph of cranial window exposing the brain for in vivo multiphoton imaging. (f) Intravital multiphoton imaging of brain of normal mice, showing diffusion of Mn@CCs (green signal) crossing BBB on normal mice. Texas Red (red signal) were pre-injected to label vasculature. g) The tracer, Evans blue, cannot permeates into the brain parenchyma in normal mice. (h) Histology of brain tissues on normal mice, confirming Mn@CCs cross the intact BBB in normal mice (n=5). Blue, nucleus; red, blood vessels; green, Mn@CCs.*

Furthermore, compare to the most used commercial gadolinium-based agent Magnevist Gd-DTPA, our agent remained in tumors far longer, and as a result was better able to discern the margins of growing tumors in orthotopic glioma and liver tumors, ever more nodules around 1 mm were easily detected and divided (new Figures 6,7,8). Visualization of the tumors in three dimensions clearly showed homogeneous distribution and high penetration in the whole tumors, even there are multinodules late-stage of intracranial tumors (Supporting video 3). This interesting property has also been rarely reported.

Fig. 6. Evaluation of Mn@CCs preferentially target the microscopic gliomas in an intracranial mouse model by MR imaging. (a,b) In vivo axial MR images of U87MG glioma-bearing mice at different time points after intravenous administration of Mn@CCs and Gd-DTPA. (c,d) Quantification of MRI signals in glioma at different time points of Mn@CCs (c) and Gd-DTPA (d) in a,b. Significant difference compared with control. * $P < 0.05$, *** $P < 0.001$. Statistical significance was assessed via Student's *t*-test. (e) Comparison of Mn@CCs and Gd-DTPA to image ability of microscopic glioma. *Significant difference compared with Mn@CCs and Gd-DTPA:

30 min ($p < 0.001$), 1 h ($p = 0.002$), 2 h ($p = 0.008$), 4 h ($p = 0.023$), respectively. $*P < 0.05$. Statistical significance was assessed via an unpaired two-tailed t -test. (f,g) *In vivo* coronal MR images of U87MG glioma-bearing mice at different time points after intravenous administration of Mn@CCs and Gd-DTPA. (h,i) Quantification of MRI signals in glioma at different time points of Mn@CCs (h) and Gd-DTPA (i) in f,g. Significant difference compared with control. $*P < 0.05$, $***P < 0.001$. Statistical significance was assessed via Student's t -test. (j) Comparison of Mn@CCs and Gd-DTPA to image ability of microscopic glioma. $*P < 0.05$. Statistical significance was assessed via an unpaired two-tailed t -test. (k) H&E staining of glioma tissue slides from nude mice. (l) *In vivo* MR images of bladder at different time points. (m) Relative signal change of bladder at different time points based on MR imaging results.

Fig. 7. 3D imaging allows visualization of the distribution and penetration of Mn@CCs to tumors. (a) T_1 -weighted MR imaging was performed before and after injection of Mn@CCs in different stages of orthotopic GBM tumor-bearing mice ($n = 3$). Day 5, Day 10, Day 20, and Day 30: The days that tumor cells were inoculated after 5, 10, 20, and 30 days to image. (b) 3D view constructed from Mn@CCs enhanced MRI of microscopic orthotopic gliomas (green color) in the brain tissues (gray color) under different angles. (c) Assessment of different early-stages different early-stages tumor volume by Amira 2020.1 (Thermo Fisher Scientific, USA) and

Clinical Eclipse Treatment Planning System (15.6). (d) Process of clearing and imaging brain and identification of small tumor (small arrows). (e) Wide-field image of whole brain from orthotopic GBM tumor-bearing mice. Gray: brain tissue, red: RFP-expressed glioma, Green: Mn@CCs.

Fig. 8. Evaluation of Mn@CCs preferentially target the microscopic and dispersive hepatocellular carcinomas in an orthotopic mouse model. (a,b) In vivo axial MR images of HepG2-bearing mice at different time points after intravenous administration of Mn@CCs (a) and Gd-DTPA (b). (c,d) Quantification of MRI signals in hepatoma at different time points of Mn@CCs (c) and Gd-DTPA (d) in a,b. Significant difference compared with control. * $P < 0.05$, * $P < 0.001$. Statistical**

significance was assessed via Student's *t*-test. (e) *Significant difference Mn@CCs compared with Gd-DTPA: 30 min ($p=0.029$), 1 h ($p=0.023$), 2 h ($p=0.003$), 4 h ($p=0.044$), 8 h ($p=0.021$), respectively. * $P<0.05$. Statistical significance was assessed via an unpaired two-tailed *t*-test. (f,g) In vivo coronal MR images of HepG2-bearing mice at different time points after intravenous administration of Mn@CCs and Gd-DTPA. (h,i) Quantification of MRI signals in hepatoma at different time points of Mn@CCs (h) and Gd-DTPA (i) in f,g. Significant difference compared with control. * $P<0.05$, *** $P<0.001$. Statistical significance was assessed via Student's *t*-test. (j) Comparison of Mn@CCs and Gd-DTPA to image ability of microscopic hepatoma. *Significant difference compared with Mn@CCs and Gd-DTPA: 30 min ($p<0.001$), 1 h ($p<0.001$), 4 h ($p=0.028$), respectively. * $P<0.05$. Statistical significance was assessed via an unpaired two-tailed *t*-test. (k) H&E staining of orthotopic hepatic tissue slides from nude mice.

In brief, all the above highlights make this work highly distinguishable from the previously reported work, which, as we believe, have obvious novelty and would have great potential impact on this field.

References:

1. J. Wahsner, E. Gale, A. Rodríguez-Rodríguez, P. Caravan. Chemistry of MRI Contrast Agents: Current Challenges and New Frontiers, **Chem. Rev.** 2019, 119, 957.
2. D. Pan, S. D. Caruthers, A. Senpan, A.H. Schmieder, S. A. Wickline, G. M. Lanza. Revisiting an old friend: manganese-based MRI contrast agents, **Wiley Interdiscip Rev Nanomed Nanobiotechnol.** 2011, 3, 162.
3. Y. Tao, L. Zhu, Y. Zhao, X. Yi, L. Zhu, F. Ge, X. Mou, L. Chen, L. Sun, K. Yang, Nano-graphene oxide-manganese dioxide nanocomposites for overcoming tumor hypoxia and enhancing cancer radioisotope therapy. **Nanoscale** 2018, 10, 5114.
4. C. Dai, Y. Chen, X. Jing, L. Xiang, D. Yang, H. Lin, Z. Liu, X. Han, R. Wu, Two-Dimensional Tantalum Carbide (MXenes) Composite Nanosheets for Multiple Imaging-Guided Photothermal Tumor Ablation. **ACS Nano** 2017, 11, 12696.
5. C. Dai, S. Zhang, Z. Liu, R. Wu, Y. Chen. Two-Dimensional Graphene Augments Nanosonosensitized Sonocatalytic Tumor Eradication. **ACS Nano** 2017, 11, 9467.
6. C Liu, Y Cao, Y Cheng, D Wang, T. Xu, L. Su, X. Zhang, H. Dong. An open source and reduce expenditure ROS generation strategy for chemodynamic/photodynamic synergistic therapy, **Nature Commun.** 2020, 11, 1735.
7. Z. Tang, Y. Liu, M. He, W. Bu. Chemodynamic Therapy: Tumour Microenvironment - Mediated Fenton and Fenton - like Reactions, **Angew. Chem. Int. Ed.** 2019, 58, 946.
8. With rising concerns over gadolinium, Anthony King looks at the alternatives under development, **Chemistry World.** <https://www.chemistryworld.com/features/new-mri-contrast-agents/1017395.article>.
9. D. Pan, A. H. Schmieder, S. A. Wickline, G. M. Lanza. Manganese-based MRI contrast agents: past, present and future, **Tetrahedron.** 2011, 67, 8431.
10. B. Drahoš, I. Lukeš, É. Tóth. Manganese(II) Complexes as Potential Contrast Agents

- for MRI, *Eur. J. Inorg. Chem.* 2012, 2012, 1975.
11. P. Fatouros, F. Corwin, Z. Chen, W. Broaddus, J. Tatum, B. Kettenmann, Z. Ge, H. Gibson, J. Russ, A. Leonard, J. Duchamp, H. Dorn. In vitro and in vivo imaging studies of a new endohedral metallofullerene nanoparticle, *Radiology*. 2006, 240, 756.
 12. Y. Liu, C. Chen, P. Qian, X. Lu, B. Sun, X. Zhang, L. Wang, X. Gao, H. Li, Z. Chen, J. Tang, W. Zhang, J. Dong, R. Bai, P. E. Lobie, Q. Wu, S. Liu, H. Zhang, F. Zhao, M. S. Wicha, T. Zhu, Y. Zhao. Gd-metallofullerenol nanomaterial as non-toxic breast cancer stem cell-specific inhibitor, *Nature Commun.* 2015, 6, 5988.
 13. W. Sun, L. Luo, Y. Feng, Y. Qiu, C. Shi, S. Meng, X. Chen, H. Chen. Gadolinium-Rose Bengal Coordinated Polymer Nanodots for MR/Fluorescence Imaging-Guided Radiation and Photodynamic Therapy, *Adv. Mater.* 2020, 32, 2000377.
 14. H. Chen, Y. Qiu, D. Ding, H. Lin, W. Sun, G. D. Wang, W. Huang, W. Zhang, D. Lee, G. Liu, J. Xie, X. Chen, Gadolinium-Encapsulated Graphene Carbon Nanotheranostics for Imaging-Guided Photodynamic Therapy, *Adv. Mater.* 2018, 30, 1802748.
 15. H. Chen, G. D. Wang, W. Tang, T. Todd, Z. Zhen, C. Tsang, K. Hekmatyar, T. Cowger, R. B. Hubbard, W. Zhang, J. Stickney, B. Shen, J. Xie. Gd-Encapsulated Carbonaceous Dots with Efficient Renal Clearance for Magnetic Resonance Imaging, *Adv. Mater.* 2014, 26, 6761.
 16. Costas D. Arvanitis, Gino B. Ferraro & Rakesh K. Jain. The blood–brain barrier and blood–tumour barrier in brain tumours and metastases, *Nature Rev. Cancer* 2020, 20, 26.
 17. E. Belykh, K. Shaffer, C. Lin, V. Byvaltsev, M. Preul, L. Chen. Blood-Brain Barrier, Blood-Brain Tumor Barrier, and Fluorescence-Guided Neurosurgical Oncology: Delivering Optical Labels to Brain Tumors, *Front Oncol.* 2020, 10, 739.
 18. J. Sarkaria, L. Hu, I. Parney, D. Pafundi, D. Brinkmann, N. Laack, C. Giannini, T. Burns, S. Kizilbash, J. Laramy, K. Swanson, T. Kaufmann, P. Brown, N. Agar, E. Galanis, J. Buckner, W. Elmquist. Is the blood–brain barrier really disrupted in all glioblastomas? A critical assessment of existing clinical data, *Neuro-Oncology*, 2018, 20, 184.

2. *Figure 2d,e: the noted r1 values in the legend do not match the slopes of the lines in the plots.*

Our response:

Thanks for the kind comment, and Figure 2d,e has been revised accordingly as follows.

Fig. 2c-f. (c) The r_1 relaxivities of Mn@CCs investigated in a 1.5 T, 7.0 T, and 9.4 T MRI system. (d) The r_1 relaxivities of glucose-Mn investigated in a 1.5 T, 7.0 T, and 9.4 T MRI system. (e) The r_1 relaxivities of Magnevist (Gd-DTPA) investigated in a 1.5 T, 7.0 T, and 9.4 T MRI system. (f) Comparison of r_1 relaxivities of Mn@CCs, glucose-Mn, and Gd-DTPA.

3. Figure 3d shows that there is some drop in survival of the cells at higher concentrations. However, the manuscript says that there is no visible toxicity which seems inconsistent. Also, a comparison with Gd-DTPA and other Mn-based MRI agents may be warranted,

Our response:

Thanks for the kind comment, and the manuscript has been revised accordingly. We compared the cytotoxicity of Gd-DTPA and Mn-based MRI agents (Mn salt, Mn-complex, and MnOx) on normal cells (NIH/3T3, LO2) and tumor cells (U87MG, HepG2).

These results indicated Mn@CCs showed less or similar toxicity than that of the safest MRI contrast agent Gd-DTPA. Under the same experiment condition, All Mn-based MRI contrast agents showed significant toxicity, including MnCl₂, MnOx, which are consistent with previous reports as cited in Ref. 18-20. These were listed in Figure 3d in the revision, and also as follows.

Fig. 3. Gliomas and hepatocyte carcinoma cell lines take up Mn@CCs, and the toxicity assessments of Mn@CCs and other Mn-based agents. (d) Cell viabilities evaluated of normal cell lines (NIH/3T3 and LO2) incubated with Mn@CCs, Gd-DTPA, and Mn-based agents for 24 h. *Significant differences from the control, $P < 0.05$. All the statistical data are expressed as mean \pm s.d. Statistical significance was assessed via a one-way ANOVA with Duncan post-hoc test.

Fig. S10. Cell viabilities evaluated by MTT assays of U87MG and HepG2 cells incubated with Mn@CCs for 24 h.

4. The manuscript claims that this agent gets past the BBB. The fluorescence results in Figure 4 seem to support this. But, then the same claim is made with the orthotopic tumor models where the BBB is already perturbed by the implanted tumor. In this case almost all MRI contrast agents will enhance the tumor.

Our response:

Thanks for the kind comment. We realized that we did not present our data very well. In our revision, we added the head captions to highlight the contents and **investigated carefully in normal health mice using MRI and multiphoton imaging, respectively.** **The new extended results in Figure 4 confirmed that Mn@CCs have the excellent ability to cross intact BBB (normal BBB) of normal mice.** However, there is almost no enhancement in MRI by Gd-DTPA under the same experiment condition.

Fig. 4. Mn@CCs cross the intact BBB in normal health mice. (a) *In vivo* MR images of normal mice at different time points after intravenous administration of Mn@CCs and Gd-DTPA. (b) Quantification of Mn@CCs MRI signals in brain parenchyma and pituitary gland of normal mice in a (n=6). No significant difference in Mn@CCs group. Statistical significance was assessed via an unpaired two-tailed t-test. (c) Quantification of Gd-DTPA in brain parenchyma and pituitary gland of normal mice in a. *Significant difference compared with the brain parenchyma and pituitary gland: P = 0.001, 0.008, and 0.0170 at 0.5, 1 and 2 h, respectively. Statistical significance was assessed via unpaired two-tailed t-test. (d) The tracer, Evans blue, cannot permeates into the brain parenchyma in normal mice. (e) Photograph of cranial window exposing the brain for *in vivo* multiphoton imaging. (f) Intravital multiphoton imaging of brain of normal mice, showing diffusion of Mn@CCs (green signal) crossing BBB on normal mice. Texas Red (red signal) were pre-injected to label vasculature. (g) The tracer, Evans blue, cannot permeates into the brain parenchyma in normal mice. (h) Histology of brain tissues on normal mice, confirming Mn@CCs cross the intact BBB in normal mice (n=5). Blue, nucleus; red, blood vessels; green, Mn@CCs.

Supplementary Fig. II. Sequential images of Mn@CCs crossing BBB on non-tumor-bearing mice. A pre-programmed program was executed every 6 min, taking 22 images in succession once time. The entire procedure was cycled 12 times for a total time of 81 min, and the first image of each circle was selected. (Scale bar: 20 μ m).

Supplementary Fig. 12. *Intravital multiphoton imaging of brain of normal mice, showing diffusion of Mn@CCs (green signal) crossing BBB on normal mice. Texas Red (red signal) were pre-injected to label vasculature.*

Then, as commented by the reviewer, it is true that the BBB is already perturbed by the implanted tumor, and the agents might enhance the tumor. However, compare to Gd-DTPA, as indicated by **Figures 6,7 (the direct comparison of Mn@CCs and Gd-DTPA)**, **Mn@CCs provided superior tumor contrast enhancement to Magnevist (Gd-DTPA) in microscopic orthotopic glioma mouse model, even at its early-stage.** The main reason might be that Gd-DTPA are nonspecific and only detect tumor masses that have significantly damaged the BBB.

Glioblastoma (GBM), as the deadliest primary brain malignancy, has a mere median survival of 14 months after diagnosis.²⁹ **One of the major reasons is lack of an effective tool for detecting glioma at its early stage when treatment is more sensitive. The blood-brain barrier (BBB) prevents 98% of small molecules and all large molecules from entering the brain.**³⁰ Currently available MRI contrast agents for the detection of brain tumors, such as Magnevist (Gd-DTPA), are nonspecific and only detect tumor masses that have significantly damaged the BBB.³¹ However, **at the early stage of glioma, they cannot be enhanced by the MRI contrast agent in most cases because the BBB has not been significantly damaged**, as indicated by a **critical assessment of**

existing clinical data.

Ref.: Jann N Sarkaria, Leland S Hu, Ian F Parney, Deanna H Pafundi, Debra H Brinkmann, Nadia N Laack, Caterina Giannini, Terence C Burns, Sani H Kizilbash, Janice K Laramy, Kristin R Swanson, Timothy J Kaufmann, Paul D Brown, Nathalie Y R Agar, Evanthia Galanis, Jan C Buckner, William F Elmquist. **Is the blood–brain barrier really disrupted in all glioblastomas? A critical assessment of existing clinical data**, Neuro Oncol. 2018, 20(2):184-191

Collectively, the superior relaxivity and imaging ability of ultrasmall orthotopic brain tumors could conclude that Mn@CCs are superior to the current clinical MRI contrast agents for orthotopic tumors and provide more reliable early-stage diagnostics.

Fig. 6. Evaluation of Mn@CCs preferentially target the microscopic gliomas in an intracranial mouse model by MR imaging. (a,b) In vivo axial MR images of U87MG glioma-bearing mice at different time points after intravenous administration of Mn@CCs and Gd-DTPA. (c,d) Quantification of MRI signals in glioma at different time points of Mn@CCs (c) and Gd-DTPA (d) in a,b. Significant difference compared with control. * $P < 0.05$, *** $P < 0.001$. Statistical significance was assessed via Student's *t*-test. (e) Comparison of Mn@CCs and Gd-DTPA to image ability of microscopic glioma. *Significant difference compared with Mn@CCs and Gd-DTPA: 30 min ($p < 0.001$), 1 h ($p = 0.002$), 2 h ($p = 0.008$), 4 h ($p = 0.023$), respectively. * $P < 0.05$. Statistical significance was assessed via an unpaired two-tailed *t*-test. (f,g) In vivo coronal MR images of U87MG glioma-bearing mice at different time points after intravenous administration of Mn@CCs and Gd-DTPA. (h,i) Quantification of MRI signals in glioma at different time points of Mn@CCs (h) and Gd-DTPA (i) in f,g. Significant difference compared with control. * $P < 0.05$, *** $P < 0.001$. Statistical significance was assessed via Student's *t*-test. (j) Comparison of Mn@CCs and Gd-DTPA to image ability of microscopic glioma. *Significant difference compared with Mn@CCs and Gd-DTPA: 30 min ($p < 0.001$), 1 h ($p < 0.001$), 2 h ($p = 0.003$), 4 h ($p = 0.042$), respectively. * $P < 0.05$. Statistical significance was assessed via an unpaired two-tailed *t*-test. (k) H&E staining of glioma tissue slides from nude mice. (l) In vivo MR images of bladder at different time points. (m) Relative signal change of bladder at different time points based on MR imaging results.

Fig. 7. 3D imaging allows visualization of the distribution and penetration of Mn@CCs to tumors. (a) T_1 -weighted MR imaging was performed before and after injection of Mn@CCs in different stages of orthotopic GBM tumor-bearing mice ($n = 3$). Day 5, Day 10, Day 20, and Day 30: The days that tumor cells were inoculated after 5, 10, 20, and 30 days to image. (b) 3D view constructed from Mn@CCs enhanced MRI of microscopic orthotopic gliomas (green color) in the brain tissues (gray color) under different angles. (c) Assessment of different early-stages different early-stages tumor volume by Amira 2020.1 (Thermo Fisher Scientific, USA) and Clinical Eclipse Treatment Planning System (15.6). (d) Process of clearing and imaging brain and identification of small tumor (small arrows). (e) Wide-field image of whole brain from orthotopic GBM tumor-bearing mice. Gray: brain tissue, red: RFP-expressed glioma, Green: Mn@CCs.

5. The *in vivo* MRI images and data in Figures 5 and 6 are reasonable. But, these figures also do not demonstrate any new capability to differentiate from existing literature. All results seem fairly typical with existing technologies.

Our response:

Thanks for the kind comment. As revised in comments 4, our Mn@CCs preferentially target these ultrasmall orthotopic tumors. Also, please see our response in comment 1 that highlight the innovative points of this article.

Our results successfully demonstrated that the Mn@CCs across an intact BBB into brain, achieving the critical first step and a key consideration in developing effective diagnostics and therapies for GBM. To the best our knowledge, there is no literature to report a potential stable and excellent MRI contrast agent than that of Gd-DTPA. In our research, the purpose is to solve the issues on safety, property, and targeting ability of the clinical MRI agents.

Collectively, the superior relaxivity and imaging ability of ultrasmall orthotopic brain tumors could conclude that Mn@CCs are superior to the current clinical MRI contrast agents for orthotopic tumors and provide more reliable early-stage diagnostics.

Fig. 6. Evaluation of Mn@CCs preferentially target the microscopic gliomas in an intracranial mouse model by MR imaging. (a,b) In vivo axial MR images of U87MG glioma-bearing mice at different time points after intravenous administration of Mn@CCs and Gd-DTPA. (c,d) Quantification of MRI signals in glioma at different time points of Mn@CCs (c) and Gd-DTPA (d) in a,b. Significant difference compared with control. * $P < 0.05$, * $P < 0.001$. Statistical significance was assessed via Student's *t*-test. (e) Comparison of Mn@CCs and Gd-DTPA to image ability of microscopic glioma. *Significant difference compared with Mn@CCs and Gd-DTPA:**

30 min ($p < 0.001$), 1 h ($p = 0.002$), 2 h ($p = 0.008$), 4 h ($p = 0.023$), respectively. $*P < 0.05$. Statistical significance was assessed via an unpaired two-tailed t -test. (f,g) *In vivo* coronal MR images of U87MG glioma-bearing mice at different time points after intravenous administration of Mn@CCs and Gd-DTPA. (h,i) Quantification of MRI signals in glioma at different time points of Mn@CCs (h) and Gd-DTPA (i) in f,g. Significant difference compared with control. $*P < 0.05$, $***P < 0.001$. Statistical significance was assessed via Student's t -test. (j) Comparison of Mn@CCs and Gd-DTPA to image ability of microscopic glioma. $*P < 0.05$. Statistical significance was assessed via an unpaired two-tailed t -test. (k) H&E staining of glioma tissue slides from nude mice. (l) *In vivo* MR images of bladder at different time points. (m) Relative signal change of bladder at different time points based on MR imaging results.

Fig. 7. 3D imaging allows visualization of the distribution and penetration of Mn@CCs to tumors. (a) T_1 -weighted MR imaging was performed before and after injection of Mn@CCs in different stages of orthotopic GBM tumor-bearing mice ($n = 3$). Day 5, Day 10, Day 20, and Day 30: The days that tumor cells were inoculated after 5, 10, 20, and 30 days to image. (b) 3D view constructed from Mn@CCs enhanced MRI of microscopic orthotopic gliomas (green color) in the brain tissues (gray color) under different angles. (c) Assessment of different early-stages different early-stages tumor volume by Amira 2020.1 (Thermo Fisher Scientific, USA) and

Clinical Eclipse Treatment Planning System (15.6). (d) Process of clearing and imaging brain and identification of small tumor (small arrows). (e) Wide-field image of whole brain from orthotopic GBM tumor-bearing mice. Gray: brain tissue, red: RFP-expressed glioma, Green: Mn@CCs.

Fig. 8. Evaluation of Mn@CCs preferentially target the microscopic and dispersive hepatocellular carcinomas in an orthotopic mouse model. (a,b) In vivo axial MR images of HepG2-bearing mice at different time points after intravenous administration of Mn@CCs (a) and Gd-DTPA (b). (c,d) Quantification of MRI signals in hepatoma at different time points of Mn@CCs (c) and Gd-DTPA (d) in a,b.

Significant difference compared with control. * $P < 0.05$, *** $P < 0.001$. Statistical significance was assessed via Student's *t*-test. (e) *Significant difference Mn@CCs compared with Gd-DTPA: 30 min ($p = 0.029$), 1 h ($p = 0.023$), 2 h ($p = 0.003$), 4 h ($p = 0.044$), 8 h ($p = 0.021$), respectively. * $P < 0.05$. Statistical significance was assessed via an unpaired two-tailed *t*-test. (f,g) In vivo coronal MR images of HepG2-bearing mice at different time points after intravenous administration of Mn@CCs and Gd-DTPA. (h,i) Quantification of MRI signals in hepatoma at different time points of Mn@CCs (h) and Gd-DTPA (i) in f,g. Significant difference compared with control. * $P < 0.05$, *** $P < 0.001$. Statistical significance was assessed via Student's *t*-test. (j) Comparison of Mn@CCs and Gd-DTPA to image ability of microscopic hepatoma. *Significant difference compared with Mn@CCs and Gd-DTPA: 30 min ($p < 0.001$), 1 h ($p < 0.001$), 4 h ($p = 0.028$), respectively. * $P < 0.05$. Statistical significance was assessed via an unpaired two-tailed *t*-test. (k) H&E staining of orthotopic hepatic tissue slides from nude mice.

6. A spelling and grammar check throughout is needed.

Our response:

Thanks for the kind comment, and our revision has been carefully read by letpub. We attached the certificate FYI.

Reviewer #2 (Remarks to the Author): Expert in brain imaging

This manuscript reports on an important advancement concerning positive contrast agents for magnetic resonance imaging (MRI). Authors synthesize and characterize carbonized complexes of Mn, showing superior performances over commercially available contrast agents.

Despite the importance of the topic and the interesting data, I think that the study, even if of high quality, does not show the potential impact necessary to be published on Nature Communications.

Our response:

We appreciate that the reviewer recognized our research is “the importance of the topic and the interesting data, I think that the study, even if of high quality”. Please see our response in comment 1 of Reviewer 1 that highlight the innovative points of this article.

Our results successfully demonstrated that the Mn@CCs across an intact BBB into brain, achieving the critical first step and a key consideration in developing effective diagnostics and therapies for GBM. To the best our knowledge, there is no literature to report a potential stable and excellent MRI contrast agent than that of Gd-DTPA. In our research, the purpose is to solve the issues on safety, property, and targeting ability of the clinical MRI agents.

Collectively, the superior relaxivity and imaging ability of ultrasmall orthotopic brain tumors could conclude that Mn@CCs are superior to the current clinical MRI contrast agents for orthotopic tumors and provide more reliable early-stage diagnostics.

1) First of all, Authors should provide better evidences about stability of their nanoparticles: presenting a Z-potential around 0 mV, I am wondering for example about long-term stability, and interactions with serum / blood proteins

Our response:

Thanks for the kind comment. We carefully investigation the stability of Mn@CCs in different media. These results demonstrated that the excellent stability of Mn@CCs.

First, our agent has long-term stability in aqueous solution.

Figure S3. Stability of Mn@CCs.

Second, our agent has excellent stability in different media. As shown in Figure S4, during the investigation, the solutions are transparent and there are almost no changes on size and zeta potential.

The surface structures of Mn@CCs effectively minimize adsorption of serum protein and blood cells (Figures S5, 6), indicating there is no interference from serum protein and blood cells during circulation, and good blood compatibility.

Supplementary Fig. 4. (a) Stability of Mn@CCs in different media, including PBS (pH 7.4), DMEM, and 10% FBS. (b-d) Size of Mn@CCs in different media at different time points. (e-g) Zeta potential of Mn@CCs in different media at different time points.

Supplementary Fig. 5. Serum protein binding test by polyacrylamide electrophoresis of Mn@CCs in PBS solution supplemented with 10% (v/v) FBS (fetal bovine serum)

for 30 min under room temperature. CBB (coomassie brilliant blue 250) was used to label FBS. The result clearly shows that the Mn@CCs have little affinity to serum protein.

Supplementary Fig. 6. (a) The interaction between Mn@CCs and blood cells was demonstrated by hemolysis test. (b) Blood-cell binding test of Mn@CCs in different concentrations (37.5, 75, 150, 300, and 600 µg/mL). The ratio of the Mn²⁺ in the plasma to the Mn²⁺ in blood cells.

2) Again concerning stability, in methods Authors are describing enzyme-like activity. Please provide explanations about the meaning.

Our response:

Thanks for the kind comment. These are employed to confirm the stability of Mn in the structure of Mn@CCs. Previously studies have revealed that the metal deposited on the surface, it reacted with H₂O₂ to generate ROS, i.e. Mn-mediated Fenton-like reaction. In our study, there is no reaction to confirm the Mn was blocked with H₂O₂. We revised this as accordingly to clear the purpose as follows:

“Confirming study that Mn²⁺ was encapsulated in carbonized shells, not deposited on the surface

To confirm Mn²⁺ was encapsulated in carbonized shells, the enzyme-like activity was measured using TMB and H₂O₂ as substrates. Typically, acetic acid-sodium acetate buffer (pH 4.0, 0.2 M) containing 10 mM H₂O₂, 416 µM TMB, and 10 µg/mL Mn@CCs were mixed thoroughly. After mixing, the reaction solution was immediately used for UV-vis spectroscopic measurements.”

3) Biocompatibility is cursorily approached, and just a short-term MTT assay on cancer cell is provided. What about effects on healthy cells?

Our response:

Thanks for the kind comment. We investigated the toxicity of Mn@CCs and other Mn-based agents using normal cell lines (NIH/3T3 and LO2) incubated with Mn@CCs, Gd-DTPA, and Mn-based agents for 24 h. As indicated in Figure 3d, compare with Gd-DTPA, Mn@CCs showed minor cytotoxicity on normal cells, such as NIH/3T3 mouse embryonic fibroblast cell and LO2 human normal liver cell (Figure 3d). Mn-based agents, including Mn salts and MnOx nanoparticles, showed significant

cytotoxicity at the same conditions due to the generation of toxic reactive oxygen species (ROS) from Mn-mediated Fenton-like mechanism (Figure 3d).

The above results (Figure 3d) and H&E staining and Serum chemistry and hematological analysis of mice treated with Mn@CCs (Figures S18), red blood cell hemolysis assay (Figures S6) indicated Mn@CCs have high biocompatibility with no leaking of Mn²⁺ ions into medium to induce cytotoxicity.

Fig. 3. Gliomas and hepatocyte carcinoma cell lines take up Mn@CCs, and the toxicity assessments of Mn@CCs and other Mn-based agents. (d) Cell viabilities evaluated of normal cell lines (NIH/3T3 and LO2) incubated with Mn@CCs, Gd-DTPA, and Mn-based agents for 24 h. *Significant differences from the control, P < 0.05. All the statistical data are expressed as mean ± s.d. Statistical significance was assessed via a one-way ANOVA with Duncan post-hoc test.

Supplementary Fig. 6. (a) The interaction between Mn@CCs and blood cells was demonstrated by hemolysis test. (b) Blood-cell binding test of Mn@CCs in different concentrations (37.5, 75, 150, 300, and 600 µg/mL). The ratio of the Mn²⁺ in the plasma to the Mn²⁺ in blood cells.

Supplementary Fig. 18. (a) H&E staining of different tissues after Mn@CCs treatment. (b) Serum chemistry and hematological analysis of mice treated with Mn@CCs. Data are mean \pm s.d.

4) Nanoparticle/cell interaction investigations are cursory as well: uptake mechanisms and long-term fate analysis are completely missing.

Our response:

Thanks for the kind comment. We investigated the nanoparticle/cell interaction, uptake mechanisms and long-term fate analysis, as follows.

There is no interference from serum protein and blood cells during circulation, and good blood compatibility (Figures S5,6).

The investigation of internalization pathway of Mn@CCs by flow cytometry showed that the cellular uptake follows the energy-dependent endocytosis and lipid raft-mediated endocytosis (Figure S9).

The accumulation analysis of the main organs showed that almost all Mn@CCs have been extracted from the body after 24 h post-injection by detecting the amounts of Mn

ions using ICP-MS (Figures 5d, S13).

We also collected and photographed the urines, which showed clear brown color in the period 1-4 h, and gradually faded in normal yellow color. Both MR results and the analysis of Mn ions in the extracted urines showed the similar trends (Figure S14).

Overall, these results confirmed that Mn@CCs can efficiently pass through the BBB and accumulate in tumors in brain tissues, and then the unbound Mn@CCs are efficiently excreted through renal clearance, which is ideal for imaging purposes.

Supplementary Fig. 5. Serum protein binding test by polyacrylamide electrophoresis of Mn@CCs in PBS solution supplemented with 10% (v/v) FBS (fetal bovine serum) for 30 min under room temperature. CBB (coomassie brilliant blue 250) was used to label FBS. The result clearly shows that the Mn@CCs have little affinity to serum protein.

Supplementary Fig. 6. (a) The interaction between Mn@CCs and blood cells was demonstrated by hemolysis test. (b) Blood-cell binding test of Mn@CCs in different concentrations (37.5, 75, 150, 300, and 600 µg/mL). The ratio of the Mn²⁺ in the

plasma to the Mn^{2+} in blood cells.

Supplementary Fig. 9. Mechanism analysis in the internalization of Mn@CCs. (a) Flow cytometry histogram analysis. (b) The corresponding proportion analysis ($n=3$) for the uptake of Mn@CCs. *Significant differences from the positive control, $P < 0.05$. ns: non-significant differences from the positive control. All the statistical data are expressed as mean \pm s.d. Statistical significance was assessed via a one-way ANOVA with Duncan post-hoc test.

Supplementary Fig. 13. Biodistribution of Mn@CCs (Mn^{2+}) in main tissues after intravenous (i.v.) administration for varied time intervals (before, 15 min, 30 min, 1, 2, 4, 8, 24, 48 and 72 h) ($n = 4$).

Supplementary Fig. 14. (a) T_1 -weighted MR images and Photograph of the urine of the healthy mice at different time points after intravenous administration of Mn@CCs. (b) Intensities of the urine of the healthy mice at different time points after intravenous administration of Mn@CCs. (c) The Mn^{2+} in the urine at different time points after intravenous administration of Mn@CCs.

5) A quantitative and long term evaluation of biodistribution is missing.

Our response:

Thanks for the kind comment. We carried out quantitative and long term evaluation of biodistribution in vivo and renal clearance as follows. these results confirmed that Mn@CCs can efficiently pass through the BBB and accumulate in tumors in brain tissues, and then the unbound Mn@CCs are efficiently excreted through renal clearance, which is ideal for imaging purposes.

Supplementary Fig. 13. Biodistribution of Mn@CCs (Mn^{2+}) in main tissues after intravenous (i.v.) administration for varied time intervals (before, 15 min, 30 min, 1, 2, 4, 8, 24, 48 and 72 h) ($n = 4$).

Supplementary Fig. 14. (a) T_1 -weighted MR images and Photograph of the urine of the healthy mice at different time points after intravenous administration of Mn@CCs.

(b) Intensities of the urine of the healthy mice at different time points after intravenous administration of Mn@CCs. (c) The Mn²⁺ in the urine at different time points after intravenous administration of Mn@CCs.

6) *Statistical analysis is not adequate. At least a post-hoc correction for multiple comparison should be performed.*

Our response:

Thanks for the kind comment. In our school (School of Public Health), there are many experts in the field of statistical analysis. We have discussed them and the new data analysis were supplied in our revision, Statistical significance was assessed via a one-way ANOVA with Duncan post-hoc test or unpaired two-tailed t-test, as indicated in Figures 3, 4, 6, 8 and supporting 9.

7) *Language needs important improvements in many points.*

Our response:

Thanks for the kind comment, and our revision has been carefully read by letpub. We attached the certificate FYI.

REVIEWERS' COMMENTS

Reviewer #1 (Remarks to the Author):

While somewhat interesting, this revised submission does not address the inherent weakness that this work is not sufficiently impactful to warrant publication in this journal. As mentioned in the original review, multiple manganese-based MRI contrast agents have been previously described and some of these agents are quite similar to the agents described in this manuscript.

The authors response to this critical comment is: "Though manganese based magnetic resonance imaging (MRI) agents (mainly Mn oxides) have been prepared before, neither the wet-chemistry synthesis approaches nor the subsequent bioimaging by the decomposition of the oxide is fully explored."

This exploration of the wet chemistry and bioimaging details would seem to make this work much more incremental rather than a significant advancement needed to warrant publication in this journal.

Reviewer #2 (Remarks to the Author):

Authors replied to most of my comments. My major concern does remain the novelty of the proposed work, that is in my opinion questionable. Said this, if editors feel the manuscript suitable for the journal I do not have objections. A further language restyling is however recommended.

Response to Reviewers

Thank you for your letter regarding the final revision for our manuscript entitled "*Carbonized paramagnetic complexes of Mn (II) as contrast agents for precise magnetic resonance imaging of sub-millimeter-sized orthotopic tumors*" (No. NCOMMS-21-10809A-Z). Revised files as well as a point-by-point response are attached in this email. The revisions have been highlighted (yellow highlight) in the revised manuscript. The detailed point-by-point responses to the reviewers' comments are listed as follows.

We sincerely hope our final revision is now acceptable into Nature Communications.

Thank you very much for handling our manuscript.

Sincerely,

Hongmin Chen, Ph.D.

Reviewer #1 (Remarks to the Author):

While somewhat interesting, this revised submission does not address the inherent weakness that this work is not sufficiently impactful to warrant publication in this journal. As mentioned in the original review, multiple manganese-based MRI contrast agents have been previously described and some of these agents are quite similar to the agents described in this manuscript. The authors response to this critical comment is: "Though manganese based magnetic resonance imaging (MRI) agents (mainly Mn oxides) have been prepared before, neither the wet-chemistry synthesis approaches nor the subsequent bioimaging by the decomposition of the oxide is fully explored." This exploration of the wet chemistry and bioimaging details would seem to make this work much more incremental rather than a significant advancement needed to warrant publication in this journal.

Our response:

Thanks for the kind comment. We have carefully finished literature research, and found that "manganese-based magnetic resonance imaging (MRI) agents (mainly Mn oxides) have been prepared before, neither the wet-chemistry synthesis approaches nor the subsequent bioimaging by the decomposition of the oxide is fully explored.^[1] In these studies, Mn oxides are degraded in tumor environment (acidic, high GSH and H₂O₂ level) to release Mn ions,^[3-5] which induce toxicity concerns as free Mn ions would generate toxic reactive oxygen species (ROS) via Fenton-like reaction.^[6-8] **The biocompatibility of manganese-based MRI agents is still a big challenge to achieve safe and efficient diagnosis via MR imaging.**^[2]

Our study successfully solves these issues. In our manuscript, carbonized paramagnetic complexes of Mn (II) are constructed that Mn²⁺ ion has to be chelated by a ligand or cage (layer). Sealed carbonized shells enhance the stability of Mn ions for high safety and contrast agent efficiency. our agent successfully crossed the intact (normal) blood–brain barrier (BBB) of health mice, confirmed by in vivo multiphoton imaging on health mice that cranial window exposes the brain. **These interesting properties have been rarely reported. In sum, our study makes this work highly distinguishable from the previously reported works, which, as we believe, have obvious novelty and would have great potential impact on this field.**

References:

1. J. Wahsner, E. Gale, A. Rodríguez-Rodríguez, P. Caravan. Chemistry of MRI Contrast Agents: Current Challenges and New Frontiers, *Chem. Rev.* 2019, 119, 957.
2. Y. Tao, L. Zhu, Y. Zhao, X. Yi, L. Zhu, F. Ge, X. Mou, L. Chen, L. Sun, K. Yang, Nano-graphene oxide-manganese dioxide nanocomposites for overcoming tumor hypoxia and enhancing cancer radioisotope therapy. *Nanoscale* 2018, 10, 5114.
3. C. Dai, Y. Chen, X. Jing, L. Xiang, D. Yang, H. Lin, Z. Liu, X. Han, R. Wu, Two-Dimensional Tantalum Carbide (MXenes) Composite Nanosheets for Multiple Imaging-Guided Photothermal Tumor Ablation. *ACS Nano* 2017, 11, 12696.
4. C. Dai, S. Zhang, Z. Liu, R. Wu, Y. Chen. Two-Dimensional Graphene Augments

- Nanosensitized Sonocatalytic Tumor Eradication. *ACS Nano* 2017, 11, 9467.
5. C Liu, Y Cao, Y Cheng, D Wang, T. Xu, L. Su, X. Zhang, H. Dong. An open source and reduce expenditure ROS generation strategy for chemodynamic/photodynamic synergistic therapy, *Nature Commun.* 2020, 11, 1735.
 6. Z. Tang, Y. Liu, M. He, W, Bu. Chemodynamic Therapy: Tumour Microenvironment - Mediated Fenton and Fenton - like Reactions, *Angew. Chem. Int. Ed.* 2019, 58, 946.
 7. With rising concerns over gadolinium, Anthony King looks at the alternatives under development, *Chemistry World.* <https://www.chemistryworld.com/features/new-mri-contrast-agents/1017395.article>.
 8. D. Pan, S. D. Caruthers, A. Senpan, A.H. Schmieder, S. A. Wickline, G. M. Lanza. Revisiting an old friend: manganese-based MRI contrast agents, *Wiley Interdiscip Rev Nanomed Nanobiotechnol.* 2011, 3, 162.

Reviewer #2 (Remarks to the Author):

Authors replied to most of my comments. My major concern does remain the novelty of the proposed work, that is in my opinion questionable. Said this, if editors feel the manuscript suitable for the journal I do not have objections. A further language restyling is however recommended.

Our response:

Thanks for the kind comment. We have carefully finished literature research, and found that "manganese-based magnetic resonance imaging (MRI) agents (mainly Mn oxides) have been prepared before, neither the wet-chemistry synthesis approaches nor the subsequent bioimaging by the decomposition of the oxide is fully explored.^[1] In these studies, Mn oxides are degraded in tumor environment (acidic, high GSH and H₂O₂ level) to release Mn ions,^[3-5] which induce toxicity concerns as free Mn ions would generate toxic reactive oxygen species (ROS) via Fenton-like reaction.^[6-8] **The biocompatibility of manganese-based MRI agents is still a big challenge to achieve safe and efficient diagnosis via MR imaging.**^[2]

Our study successfully solves these issues. In our manuscript, carbonized paramagnetic complexes of Mn (II) are constructed that Mn²⁺ ion has to be chelated by a ligand or cage (layer). Sealed carbonized shells enhance the stability of Mn ions for high safety and contrast agent efficiency. our agent successfully crossed the intact (normal) blood–brain barrier (BBB) of health mice, confirmed by in vivo multiphoton imaging on health mice that cranial window exposes the brain. **These interesting properties have been rarely reported. In sum, our study makes this work highly distinguishable from the previously reported works, which, as we believe, have obvious novelty and would have great potential impact on this field.**

References:

1. J. Wahsner, E. Gale, A. Rodríguez-Rodríguez, P. Caravan. Chemistry of MRI Contrast Agents: Current Challenges and New Frontiers, *Chem. Rev.* 2019, 119, 957.

2. Y. Tao, L. Zhu, Y. Zhao, X. Yi, L. Zhu, F. Ge, X. Mou, L. Chen, L. Sun, K. Yang, Nano-graphene oxide-manganese dioxide nanocomposites for overcoming tumor hypoxia and enhancing cancer radioisotope therapy. *Nanoscale* 2018, 10, 5114.
3. C. Dai, Y. Chen, X. Jing, L. Xiang, D. Yang, H. Lin, Z. Liu, X. Han, R. Wu, Two-Dimensional Tantalum Carbide (MXenes) Composite Nanosheets for Multiple Imaging-Guided Photothermal Tumor Ablation. *ACS Nano* 2017, 11, 12696.
4. C. Dai, S. Zhang, Z. Liu, R. Wu, Y. Chen. Two-Dimensional Graphene Augments Nanosonosensitized Sonocatalytic Tumor Eradication. *ACS Nano* 2017, 11, 9467.
5. C Liu, Y Cao, Y Cheng, D Wang, T. Xu, L. Su, X. Zhang, H. Dong. An open source and reduce expenditure ROS generation strategy for chemodynamic/photodynamic synergistic therapy, *Nature Commun.* 2020, 11, 1735.
6. Z. Tang, Y. Liu, M. He, W, Bu. Chemodynamic Therapy: Tumour Microenvironment - Mediated Fenton and Fenton - like Reactions, *Angew. Chem. Int. Ed.* 2019, 58, 946.
7. With rising concerns over gadolinium, Anthony King looks at the alternatives under development, *Chemistry World.* <https://www.chemistryworld.com/features/new-mri-contrast-agents/1017395.article>.
8. D. Pan, S. D. Caruthers, A. Senpan, A.H. Schmieder, S. A. Wickline, G. M. Lanza. Revisiting an old friend: manganese-based MRI contrast agents, *Wiley Interdiscip Rev Nanomed Nanobiotechnol.* 2011, 3, 162.

Our revision has been carefully read by letpub again.